# Attractor dynamics with activity-dependent plasticity capture human working memory across time scales

Connor Brennan[1] & Alex Proekt [1✉]

Most cognitive functions require the brain to maintain immediately preceding stimuli in working memory. Here, using a human working memory task with multiple delays, we test the hypothesis that working memories are stored in a discrete set of stable neuronal activity configurations called attractors. We show that while discrete attractor dynamics can approximate working memory on a single time scale, they fail to generalize across multiple timescales. This failure occurs because at longer delay intervals the responses contain more information about the stimuli than can be stored in a discrete attractor model. We present a modeling approach that combines discrete attractor dynamics with activity-dependent plasticity. This model successfully generalizes across all timescales and correctly predicts intertrial interactions. Thus, our findings suggest that discrete attractor dynamics are insufficient to model working memory and that activity-dependent plasticity improves durability of information storage in attractor systems.

[1] University of Pennsylvania, 3160 Chestnut St., Philadelphia, PA, USA.  ✉email: proekt@gmail.com

Working memory—a flexible limited capacity system which temporarily maintains and stores salient information in the absence of sensory cues—is an essential requirement for most human cognitive processes[1,2]. Despite its fundamental importance, working memory is unreliable. Representations stored in working memory degrade over time[3–6]. Some aspects of memory decay are random. In the setting of spatial working memory for instance, this form of memory decay gives rise to the random dispersion of recalled locations relative to the original stimulus[7]. Furthermore, working memory in different domains is systematically biased[8–12]. For instance, recalled spatial locations[8,9,12] and colors[10,11] systematically deviate from the original stimuli. Much like the random errors, biases in working memory also grow over time[10,13].

Persistent neuronal firing[14–17] observed during the delay period between stimulus and response in the prefrontal cortex led to the hypothesis that working memory trace is maintained by a stable "attractor" state established through recurrent excitation[18,19]. Such stabilization of neuronal activity may allow the neuronal network to store sensory information in the absence of the stimulus and counteract the effect of noise—ubiquitous in most aspects of brain physiology[20–22].

While the importance of attractors in memory storage has received considerable experimental support[16,23,24], two competing attractor models have been proposed. Continuous attractor models propose that memories are stored by a continuum of stable states such as line or ring attractor[19,23–25] particularly well suited for remembering continuous variables such as location in space. Alternatively, it has been proposed that in order to counteract the effect of noise[26], discrete attractors are used to store stimuli in working memory[10,27–29]. In contrast to continuous attractor models that give rise to unbiased memory storage, discrete attractor models can be fit to reproduce experimentally observed biases in working memory and can be shown to have noise-stabilizing properties[10]. Despite some significant differences between the predictions made by the continuous and the discrete attractor models, there is a fundamental assumption that is common to both formulations. Specifically, both classes of models assume that the dynamics—the laws of motion that govern how neuronal activity changes during the delay between the stimulus and its recall—are fixed.

Under this assumption, the dynamics of the memory trace can be modeled as drift and diffusion on a fixed energy landscape. For a continuous line or ring attractor network, the energy landscape has a linear or a circular energy trough; while for a discrete attractor network, the energy landscape has discrete energy wells[30]. Local minima in the energy landscape serve to stabilize neuronal activity against noise and other perturbations and can give rise to sustained neuronal activity in the absence of stimulus. This energy landscape approach has been recently used to provide evidence that working memory in humans and non-human primates is mediated by discrete attractors[10].

An example of the drift-diffusion model used to model working memory is shown in Fig. 1A, B. A participant is presented with the stimulus (e.g., location of a dot along a line) and is then asked to recall the stimulus location after some delay. During a delay period between the stimulus and the response, the memory trace evolves under the influence of two classes of forces. Diffusion adds uncorrelated noise which accumulates over time. This results in the random unbiased dispersion of the recalled location relative to the target stimulus such that in the limit of long delays, the responses tend towards a uniform distribution. This is the expected long-term behavior of a continuous attractor model corrupted by noise[26]. To mitigate this damaging effect of noise, it has been proposed that in addition to diffusion, drift towards a local minimum of the energy landscape serves to stabilize the memory trace. Under this discrete attractor model, the memory trace diffuses down the energy gradient towards a local minimum. Once at the local minimum, the concave shape of the energy landscape serves to stabilize the memory trace against the corrupting influence of noise. After the memory trace has had sufficient time to relax towards the local energy minimum, the distribution of responses is determined by the shape of the energy landscape—the probability of observing a response at a specific location along the target interval is related to the energy through the Boltzmann equation. Consistent with the predictions of the discrete attractor model, it has been shown that, given uniformly distributed target stimuli, the distribution of responses exhibits several discrete peaks corresponding to the local minima of the energy landscape[10].

However, a key prediction of the discrete attractor model was not explicitly tested. Specifically, the discrete attractor model predicts that after the memory trace relaxes to the energy minimum and the biases saturate, the only information that can be recovered from the response is the location of its nearest energy minimum. In other words, the maximal mutual information between the stimulus and the response for a discrete attractor model is solely limited by the number of discrete energy wells in the system. Here, we test this prediction by attempting to approximate human visual working memory performance using drift diffusion models.

## Methods

**Participants**. 161 human participants engaged in an online experiment administered through Prolific. We screened for attention to the task by removing trials with a bias of greater than 0.25 (210 out of 17760 trials). This threshold is set at roughly the 99% quantile of all recorded biases to avoid outlier artifacts. Each participant was asked to complete at least 100 trials at two different delays. They were then given the option to complete more trials at a new set of two delays for increased reward. Data reported in the main text uses the aggregate dataset pooled across participants.

The average age of participants was 26.2 with a standard deviation of 8.6. Information about the participants' sex was obtained through self-report. There were 86 male participants, and 66 female participants. Nine participants chose not to answer.

All experiments in this study were approved by Institutional Review Board at the University of Pennsylvania and were conducted in accordance with the National Institutes of Health guidelines. All participants gave written informed consent before beginning the experiment.

The study was not preregistered.

Participant compensation was described as follows:

"The task is divided into blocks of 10 trials each. You will be asked to complete at least 10 blocks. You will be compensated at a rate of $10.00 USD an hour if you complete 10 blocks. If you choose to complete more blocks your compensation will increase by $1.00 USD per hour for every 10 blocks completed."

**Experiment**. The experiment web app was developed in Unity. During the experiment participants were asked to recall the position of a dot on a line after a delay of 0, 1, 3, 6, 13 or 20 s. The target position of the dot along the line was drawn randomly on each trial from a uniform distribution between 10 and 90% along the length of the line to avoid edge effects. The size of the target dot had a diameter of 15 pixels and the length of the line was 200px.

Participants were first asked to complete a short survey, followed by an example trial where the instructions were printed on each slide and the slides only advanced when the participant

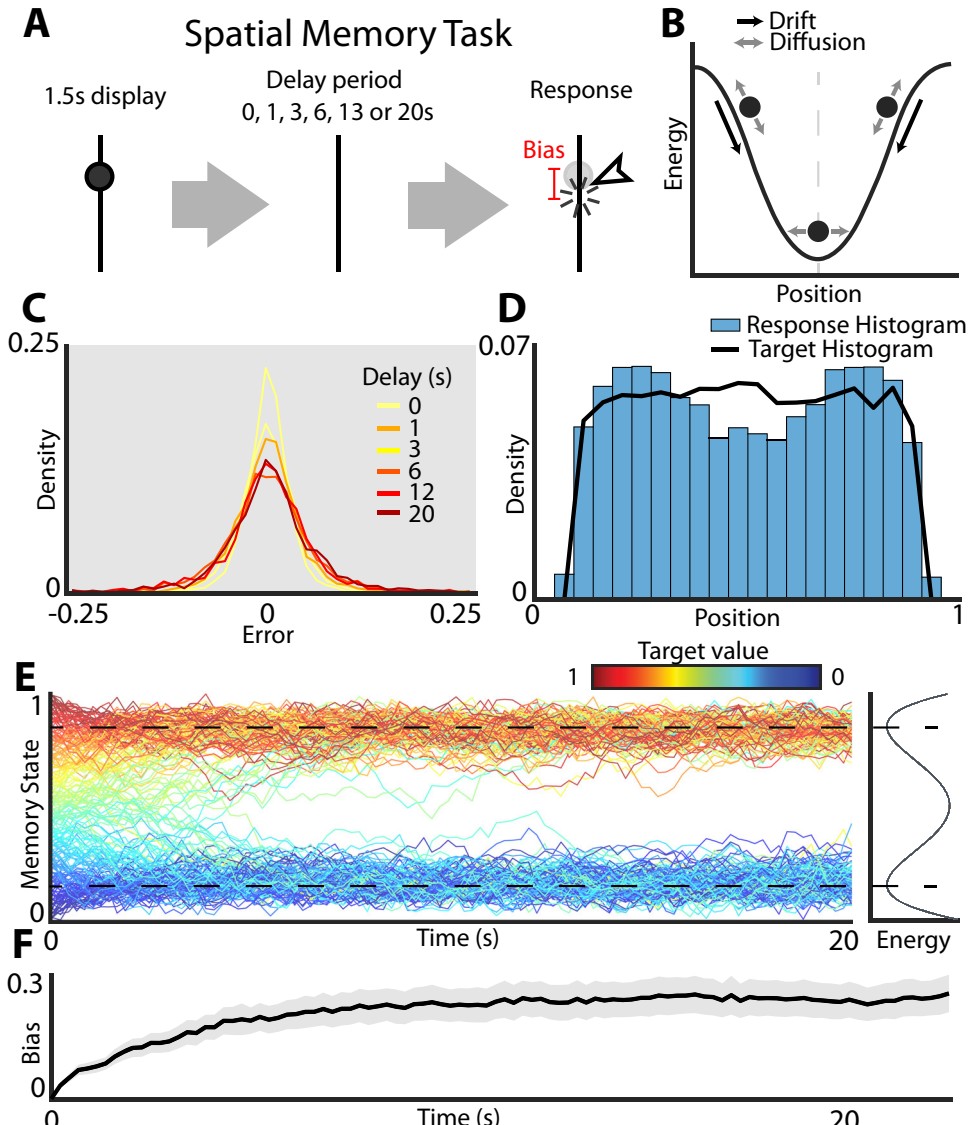

**Fig. 1 Schematic of the energy landscape model of visual working memory. A** Experimental design. Participants are asked to remember the position of a dot on a line. After a variable delay period, the participants are asked to indicate the remembered location of the dot with a mouse click. **B** Schematic of diffusion on an energy landscape used to model working memory dynamics. The dynamics of the memory trace are governed by two components: drift (black arrows) given by the local gradient of the energy landscape (black curves) and diffusion (gray arrows) which gives rise to random fluctuations of the memory trace. Under the influence of both drift and diffusion, the memory trace will over time move to the local energy minimum and randomly fluctuate around the minimum (vertical gray dashed line) under the influence of noise. **C** Distribution of errors at different delays. Errors were calculated as Error = Response value - Target value across the entire number line (targets ranging from 0.1 to 0.9). **D** Histogram shows the density of responses. The black line shows the distribution of the targets. Targets were drawn from a uniform distribution between 0.1 and 0.9 to avoid edge effects. **E** Simulation of two well energy landscape (curve on the right margin) illustrates the dynamics of working memory. Color indicates starting value of the trace (location of the target stimulus). Note that all traces eventually converge on the bottom of the energy wells (dashed black lines) at long timescales. **F** As the memory trace evolves down the energy gradient, the memory becomes systematically biased. The time course of development of such biases (averaged across all locations) is shown by the black trace (shaded area shows 95% confidence interval). Note that bias saturation occurs when the system approaches the bottom of the energy well because at this point the drift is negligible and the system becomes purely diffusive.

clicked. The first slide showed a line with a dot on it and the text "You must remember the position of the slider on the line. Click anywhere to continue. (Next time this will auto-advance) Your cursor has been hidden, please don't use your finger." The next slide was the delay period in which a blank slide with the text "You will wait a few seconds before being asked to respond." was displayed for 3 s (in future trials the display time of this slide was randomized). Finally, the response slide was displayed showing the line in the same position but without the target dot. A skip button in the button right of the screen was added to allow

participants to skip a trial. In the example trial the following instruction was given as text "Click or drag the slider to the original position. The closer the better! If you're not sure, press skip for no penalty." Once the participant clicks on the line the true experiment starts.

During the execution of the task, the participant first sees a fixation slide that has the text "Get ready!" in the middle of the screen. Next, they see the target slide with the dot for 1.5 s. Then the blank slide is shown for a variable delay interval. During the delay interval, the cursor disappeared and appeared in the middle

of the screen upon conclusion of the delay. Each participant saw 2 of the 6 possible delay times (0, 1, 3, 6, 13, and 20 s). The delay was chosen randomly at the beginning of the experiment and every 100 trials (10 blocks). The experiment was divided into two separate runs. The first run used only the delay time (0, 1, 3, and 6 s) while the second run used only the delay times (1, 13, and 20 s). Finally, the participants return to the response screen with the blank line and the skip button. This process repeats for a 10-trial block. Once a block is finished the participant receives the text "Good job! Click anywhere to start another 10 trials." This process repeated for at least 100 trials (10 blocks) with the option to do more for extra reward. There was no time limit for the response.

**Effect of delay on bias.** The error of each trial was defined as the difference between the target and the response value. Bias of responses at a given position along the line was then calculated by binning each trial into non overlapping bins based on target value with a width of 0.25 (41 total bins along the line). The bias of each bin was defined as the mean error for all trials belonging to that bin. To calculate the overall bias for a given delay time we took the maximum absolute bin bias across all bins associated with that delay. This process was then bootstrapped 1000 times to estimate the variance.

Model goodness of fit is reported as coefficient of determination ($R^2$) and earth mover's distance (EMD). $R^2$ was computed on a vectorized version of the distributions. For computational efficiency, EMD was calculated individually for each target location and then averaged across target locations. We implemented EMD using a MATLAB function[31] based on Ruber et al. [32].

To characterize the time constant that governs the development of the overall bias, the following exponential function was fit to the data using constrained minimization implemented via the MATLAB function fminsearch:

$$B(t) = \eta\left(1 - e^{-\lambda t}\right) + B_0 \quad (1)$$

Where $\eta + B_0$ is the overall bias observed at saturation, $\lambda$ is the time constant, and $B_0$ is the bias estimate for the initial delay time. The fit was optimized using a least squares cost function between the expected bias, $B(t)$, and the observed bias for each timepoint in the data. The fit was bootstrapped 100 times to estimate the variance of the time constant.

**Effect of delay on mutual information.** Mutual information of the target value and participant response was calculated using 12 non-overlapping bins along the line to support the distributions. The marginal distributions of trial targets, $p(T)$, and participant response, $p(R)$, were calculated by normalizing the trial count in each bin for the respective variable. The joint distribution, $p(T, R)$, was calculated using normalized trial counts in a 2D grid with the same bin sizes as the previous distributions. Mutual information was then calculated as:

$$MI(T, R) = \sum p(T, R) \ln \frac{p(T, R)}{p(T)p(R)} \quad (2)$$

Where the sum is taken over all bins, and elements in which $p(T)p(R) = 0$ were set to 0. The value was calculated independently for each delay time and bootstrapped 100 times to estimate the variance.

**Static landscape model definition.** The dynamics of spatial memory were first modeled as Brownian motion on a fixed energy landscape as in ref. [10]:

$$dx = \left(\beta G(x) + \varepsilon(\sigma)\right)dt \quad (3)$$

Where $x$ is the position of the system along the line, $\beta$ is a scalar encoding drift strength, and $G(x)$ is the amount of drift at each location along the line and corresponds to the negative gradient of the energy landscape, $G(x) = -\frac{dE(x)}{dx}$. $\varepsilon(\sigma)$ is zero mean Gaussian noise with variance of $\sigma$ and $dt$ is the time step.

**Plastic landscape model definition.** Plasticity was modeled as a local deformation of the energy landscape. Specifically, we hypothesized that in order to explain the unexpectedly high amount of mutual information between the stimulus and response present at a steady state, the memory trace must be locally stabilized. This stabilization was modeled as a Gaussian depression added to the landscape at each step. This depression was centered on the current position of the system, and its amplitude and width are parameters of the model fit to the experimental observations (see below):

$$G(x, t + 1) = G(x, t) + \zeta_P(x)\beta_P dt \quad (4)$$

Where $G(x, t)$ is the drift of the system at location x and time $t$, $\zeta_P(x)$ is the plastic kernel and $\beta_P$ is the scalar which encodes the growth rate of the local deformation of the energy landscape. Because, the drift of the memory trace is given by the negative derivative of the energy landscape, the plastic kernel, $\zeta_P(x)$, is defined as the derivative of a Gaussian centered at the present location of the memory trace $x_0$ with parameter plastic sigma, $\sigma_P$:

$$\zeta_P(x) = -(x_0 - x)e^{-\frac{(x_0-x)^2}{2\sigma_{P^2}}} \quad (5)$$

Note that the positive derivative gives a negative Gaussian on the energy landscape. Furthermore, note that if the plasticity time constant, $\beta_p$, is set to zero Eq. (4), describes a static energy landscape and the dynamics of the system are reduced to those in Eq. (3). Thus, the plastic landscape model introduces two additional parameters, $\beta_P$ and $\sigma_P$ which govern the development and the shape of the local deformation of the energy landscape. Both the static and the plastic landscape models were fit to the data using identical procedure described below.

**Fitting the energy landscape models to data.** $G(x)$ was fit at 50 equidistant bins spanning the length of the line. The value of $G(x)$ at each location is a parameter of the model. To reduce dependence of the model on a single value of $G(x)$, the drift was smoothed using a Guassian kernel with variance of 0.2.

We fit the model to data collected at six delay times and sampled uniformly over the range of target values between 0.1 and 0.9. In order to synchronize the time course of the model dynamics and observed data we include a delay term, $t_0$, into the model such that:

$$t_{\text{fit}} = t_0 + t \quad (6)$$

Where $t_{\text{fit}}$ is the time used to fit the data and $t$ is the simulation time. We initialized the model at 50 starting positions ranging from 0.1 to 0.9 and simulated 300 runs of the model (Eq. (3)) for each of these start points. The $dt$ of the model was fixed to 0.1 and the model was run until $t_{\text{fit}}$ reached 20 s. This occurs after $\frac{20+t_0}{dt}$ time steps. This implicitly assumes that the dynamics of the memory trace on the interval $[0, t_0]$ are identical to those observed subsequently. In additional model simulations, we relaxed this assumption and allowed the drift constant $\beta$ and noise variance $\sigma$ to assume different value during the time up to $t_0$. This introduced two new parameters to the model ($\beta_{t_o}$ and $\sigma_{t_o}$), but otherwise the fitting procedure was unchanged.

The models were fit to approximate the distribution of responses at all target locations and delay intervals to ensure that both bias and variance were well approximated. The data

**Table 1 Model parameters and value ranges.**

| Parameter name | Lower bound | Upper bound | Static fit | Static meta model | Plastic fit |
|---|---|---|---|---|---|
| Drift, $G(x)$ | −5 | 5 | N/A | N/A | N/A |
| Drift strength, $\beta$ | 0.03 | 0.2 | 0.0481 | 0.0998 | 0.0794 |
| Noise strength, $\sigma$ | 0 | 0.2 | 0.0821 | 0.0759 | 0.0977 |
| Delay term, $t_0$ | 1.0 | 4 | 1.22 | 3.50 | 2.4843 |
| Plastic weight, $\beta_P$ | 0.1 | 15 | N/A | N/A | 9.9976 |
| Plastic sigma, $\sigma_P$ | 0.015 | 0.05 | N/A | N/A | 0.0218 |

List of all fit parameters for models mentioned in the Main Text. Abbreviations are the same as used in the Main Text.

distribution was calculated by binning the trials by target value into 50 bins from 0.1 to 0.9 at each delay interval. For each target bin, the distribution of responses was estimated using MATLAB's ksdensity function with 50 bins between −0.25 and 0.25 relative to the target value as support. The modeled distributions of responses were calculated in the same way for each of the 50 target locations used. In order to reduce the effect of the stochastic nature of the model, we smoothed the model distributions using a moving mean filter with a stencil size of 3 before comparing to the data distributions. Kullback–Leibler divergence (KL divergence) between model and experimentally observed distributions summed over all six delays was used as the cost function.

Model optimization was performed using the pattern search algorithm in MATLAB. We first optimized the full parameter set (50 drift parameters, $G(x)$ that define the gradient of the energy landscape, and the parameters that set the dynamics of the system on this energy landscape $\beta$, $\beta_p$, $\sigma$, $\sigma_p$, and $t_0$), and then in a second run optimized just the dynamical system parameters while keeping $G(x)$ fixed. For static landscape model $\beta_p$ and $\sigma_p$ were set to zero. See Table 1 for a full list of the range of parameter values explored.

**Static landscape model fit to optimize the development of bias.** Instead of directly matching the distributions of the data we attempted to produce a static landscape model that matched just the delay time vs. bias curves. The model and simulation are the same as used for the static landscape model. The only modification was that the cost function was changed to be the squared error of the biases observed in the data compared to the biases given by the model at each time point and the squared error of the model's fit time constant and the observed time constant. These two terms were normalized by dividing by the observed value:

$$C = \frac{(B_{model} - B_{data})^2}{B_{data}} + \frac{(\tau_{model} - \tau_{data})^2}{\tau_{data}} \qquad (7)$$

Where $C$ is the cost function, B is the bias of at a given time delay defined as the maximum absolute value of the mean error across spatial bins, and $\tau$ is the time constant of the fit exponential model.

**Inter-trial effects.** In order to calculate the spatial dependence of inter-trial effects we first binned the data based on the current trial's target position and the delay time. We used 50 non overlapping bins from 0.1 to 0.9 with width 0.1 to spatially partition the current trials. We then calculated the error for different previous trial responses relative to the current trial's target by iterating through 21 offsets from −0.3 to 0.3. For each offset we estimated the error using a weighted average of all trial errors in the target bin and weighted by the offset of the previous trial's response. The weighting used a Gaussian kernel with variance of

0.05. Mean and confidence intervals are estimated by bootstrapping the trials used in the calculation 100 times.

The temporal dependence of inter-trial effect was calculated by taking a stereotypical spatial bin and calculating the spatial dependence for this spatial bin at each of the observed delay times. The total size of the effect at each time point is calculated by taking the range (maximum value - minimum value) of the spatial effect. Mean and confidence intervals were estimated by bootstrapping the calculation 100 times.

Finally, in order to test the level of inter-trial effect predicted by the plastic model, we add a term that allows the plasticity in the previous trial to decay during the inter-trial period.

$$G(x,0) = \lambda_T G_0(x) + (1 - \lambda_T) G_f(x) \qquad (8)$$

Where $G_0(x)$ is the static landscape given by the model parameters, $\lambda_T$ is the plastic trial decay value and $G_f(x)$ is the final landscape in the previous trial. This value was not explicitly fit to the data as the trials were self-paced. Instead different values of $\lambda_T$ were explored to examine the possible range of intertrial interactions that can be produced by the plastic landscape model.

**AIC and BIC calculations.** All models are fit to approximate the distribution of responses at all target locations and delay intervals. Thus, we can explicitly compute the log likelihood of the model given the data by taking the value of the normalized bin for a given response, target, and delay tuple. AIC and BIC were then calculated using the number of parameters in the model $k$, the likelihood $\hat{L}$ and the total number of responses $n$:

$$AIC = 2k - 2\ln(\hat{L}) \qquad (9)$$

$$BIC = k\ln(n) - 2\ln(\hat{L}) \qquad (10)$$

Finally, the AIC and BIC $p$ values were calculated by bootstrapping the AIC and BIC values of both the plastic and static models and calculating the number of times the plastic model gave a higher AIC or BIC than the static model. Note that a $p$-value of <0.001 means that no bootstrapped pairs corresponded to the null hypothesis that the values overlap.

**Statistics.** $p$-values were calculated empirically from the bootstrapped data using resampling with replacement. This method ensures that even if we were to increase the sample size of our simulations, we will not artificially inflate the power of our statistical tests. The dataset had 17550 total trials, and simulations were run 1000 times to be able to detect $p$-value of up to 0.001.

For $p$-values of model time constants vs. the data time constant we first found the distributions of the model time constants, and then calculated the quantile of this model distribution that was greater than the mean value observed in the data. This same method was used to calculate the $p$-value that the mutual information in the data and plastic model were greater than the results from the static model.

By making use of this empirical permutation testing, we avoid needing to make any assumptions about the distribution of the data, and instead rely only on empirical estimates of the distributions of our measures of interest (i.e., time constant, AIC value, etc.)

**Plastic fit robustness**. We performed a robustness study on the fit parameters of the plastic model. Starting from the parameter values given in the main paper we systematically varied the Plastic weight and Plastic sigma parameters over the full search range defined during fitting. Note that due to computational constraints we opted to vary each of these two parameters separately. We then ran the loss function used for fitting on the new parameter values a total of 5 times and reported the mean and standard deviation of the loss.

**Robustness of fitting procedure**. In order to verify the validity of our fitting procedure we fit simulated data. Simulations were performed with an idealized two well energy landscape at varying levels of noise and drift parameters. Parameters were drawn from a mesh grid of 10 noise and 10 drift parameters, for 100 total combinations. The 10 noise and drift values were chosen to be in the range of parameter values sampled in the main paper. Each combination of parameters was simulated 50 times from 50 equally spaced starting positions and sampled at the same time intervals as our dataset. This resulted in 1500 samples for the fitting procedure, which is about an order of magnitude less data than available in our dataset. Due to the large number of fits, we opted to fix the energy landscape during fitting.

**Reporting summary**. Further information on research design is available in the Nature Portfolio Reporting Summary linked to this article.

## Results
**Energy landscape models of working memory**. Here, our primary goal was to model the performance of human volunteers (161 participants, 17550 trials) on a simple spatial working memory task. The task consisted of an initial fixation (1.5 s) followed by a presentation of a line segment with a dot placed at a random location along the line. After a variable delay (Methods), the participants were instructed to indicate with a mouse click where they thought the dot was located (Fig. 1A). The response bias was defined similarly to previous work as the deviation of the mean of recalled locations from the target location. The presence of such biases has been well documented in a number of working memory tasks[8–11]. Accurate estimation of these biases at each location over multiple delays requires more trials than are available at the level of an individual participant. Thus, in what follows we aggregate the data across all 161 participants. This aggregation is justified as the distribution of biases observed in individual participants is similar across participants (Supplementary Fig. 1).

A general class of models used to explain the development of errors and biases cast the dynamics of working memory as diffusion on an energy landscape (Fig. 1B). The memory state starts out near the target location and then evolves during the delay period under the influence of two forces: drift and diffusion. Diffusion is modeled as a random walk and gives rise to the random dispersion of recalled locations relative to the target. The experimentally observed dispersions of recalled location as a function of the delay period is shown in Fig. 1C. As expected from the accumulation of random noise, the dispersion of responses grows with increasing delay period.

In contrast to diffusion which does not have a preferred direction, drift always points in the direction of the local energy minimum which corresponds to an attractor state of the network. Thus, the shape of the energy landscape deforms the overall distribution of responses. As a consequence of drift down the energy gradient, responses accumulate near the bottom of the wells. Consistent with this model, given the uniform distribution of target stimuli, the distribution of responses exhibits two clear peaks (Fig. 1D). This suggests that the energy landscape governing the dynamics of working memory in this task has two wells. To better visualize how these peaks develop, Fig. 1E shows simulation of memory traces on a hypothetical energy landscape that contains two wells separated by an energy maximum. While target locations are uniformly distributed on a line segment (colored traces), over time the responses gravitate towards the energy minima of the landscape giving rise to a highly non-uniform distribution of responses. This model naturally gives rise to biased memory storage. Note that if the target location is presented near the energy maximum, the memory traces systematically drift away from the target over time. The direction of this drift depends on whether the target was just above or just below the local energy maximum. The development of bias given by this hypothetical two well energy landscape is shown in Fig. 1F. Bias grows over time and eventually saturates when the memory trace approaches the bottom of the energy well. At steady state, when the system has sufficient time to decay towards its corresponding energy minimum, all drift subsides and the dynamics become purely diffusive. In what follows, we will explicitly fit an energy landscape model to working memory performance to determine whether it can in fact reproduce the distribution of responses observed in human volunteers performing a visual working memory task in Fig. 1A.

**Energy landscape models fit to human working memory data fail to recapitulate the dynamics of working memory**. Consistent with previous findings[13], experimental results revealed that the bias of spatial working memory grew over time, eventually reaching a plateau (Fig. 2A). This behavior is qualitatively similar to that found for a hypothetical energy landscape with two wells (Fig. 1F). The responses systematically deviated from the target locations consistent with an energy landscape with discrete energy wells (Fig. 2C top). These deviations were exacerbated with longer delays (Fig. 2D top). The general shape of these deviations of responses (spline through the distributions) informs the general shape of the energy landscape. Locations where this curve crosses zero, and unbiased responses are obtained, correspond to the extrema of the energy landscape. When the slope of the line is positive, any deviations from the extrema become exacerbated by the bias (black circles in Fig. 2C, D). These unstable points correspond to the energy maxima. Conversely, zero crossings with negative slope are stable because all perturbations are opposed by the drift (red circles Fig. 2C, D). These stable points correspond to energy minima. Thus, the shape of the response distribution (Fig. 1D) and the systematic deviation of responses (Fig. 2C, D) are both consistent with the diffusion on an energy landscape with two energy wells separated by an energy peak located close to the middle of the line. These results are also similar to those in Panichello et al.[10] who used a very similar approach to model performance on a visual working memory task which used colors rather than spatial locations.

While saturating biases and the probability distributions of responses (Fig. 2C, D top) are qualitatively consistent with diffusion on an energy landscape with two wells, it is not *a priori* obvious that the dynamics of the development of such bias across

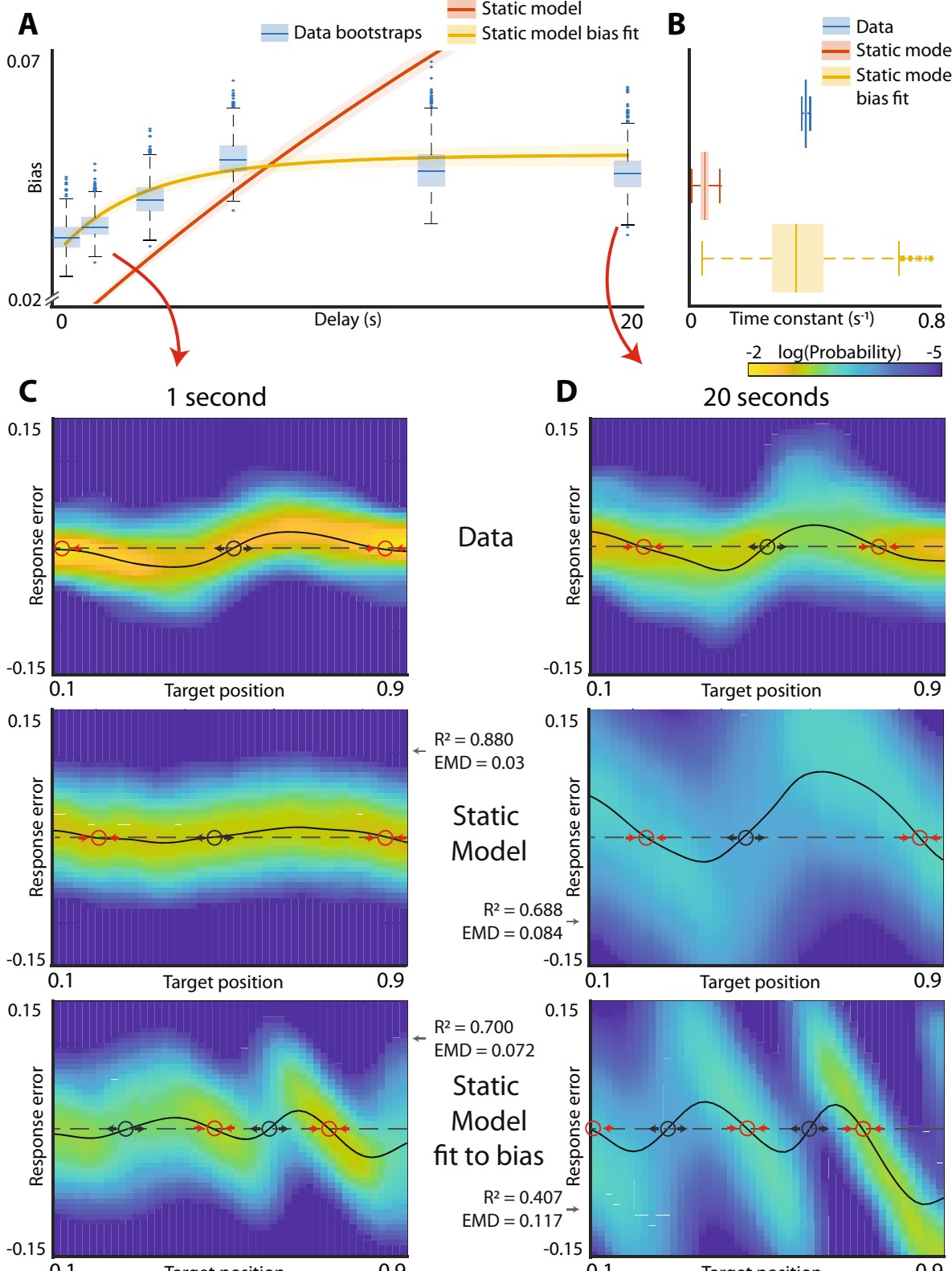

**Fig. 2 Experimental observations and energy landscape model fits. A** Development of bias in the visual working memory task. Blue boxplots are individual bootstrapped estimates of bias calculated from human participants. Orange line and shaded area are the mean bias and 95% confidence interval for the energy landscape model fit to the raw data. Yellow line is the bias of the energy landscape model fit explicitly to approximate the experimentally observed development of bias. **B** Distributions of time constants of the bias saturation curves. **C** Response distributions after 1 s delay for experimental data (top), energy landscape model fit to raw data (middle) and energy landscape model fit to bias curve (bottom). Red circles denote stable fixed points in the landscape. Black circles denote unstable fixed points. Reconstruction $R^2$ values and earth mover's distances (EMD) are reported for model fits. **D** Same as (**C**) but after 20 s delay.

different delay periods can be faithfully fit by diffusion on such landscape. To determine whether this is the case, using methods similar to that used by Panichello et al.[10], we fit the entirety of the behavioral data to an energy landscape model. In a slight departure from previous attempts, our modeling approached was based on Monte Carlo methods and was aimed at approximating the distribution of responses observed at every target position and time delay. This was accomplished by minimizing the Kullback-Leibler divergence between experimentally observed and modeled distributions (Methods). The validation of this modeling approach is shown in Supplementary Fig. 2.

As expected from qualitative observations of the data (Fig. 1D, Fig. 2C, D), the fit of the data yielded an energy landscape with two potential wells (Supplementary Fig. 3). As expected, this model gave rise to progressive increase (Fig. 2A red) and eventual saturation of biases (Supplementary Fig. 4). The rate of bias development in the model, however, was clearly different from experimental results (Fig. 2A). Consistent with results in Fig. 2A, the time constant for bias development in the model was approximately an order of magnitude smaller than in experimental observations (Fig. 2B, $p$ value < 0.001). The energy landscape model faithfully fit the distribution of responses at short delays (Fig. 2C middle row, R^2 = 0.880, earth mover's distance (EMD) = 0.03) but deviated from the data at longer delays (Fig. 2D middle row, $R^2$ = 0.688, EMD = 0.084). Previous work on modeling working memory performance using drift-diffusion models separated the overall dynamics of the system into two components: "encoding" and "memory"[10]. To determine whether this change in the modeling approach can rescue the performance of the drift-diffusion model, we allowed the drift strength and noise during the initial "encoding" period to be distinct from those observed subsequently (Methods). This, however, did not improve model performance (Supplementary Fig. 5). Thus, while the qualitative features of working memory are similar to those expected for diffusion on an energy landscape, a drift-diffusion model does not faithfully fit experimental observations across multiple delays.

To determine whether an attractor system can be used to approximate the dynamics of development of bias, we fit the static landscape model constrained only by the dynamics of development of biases and agnostic to the actual distribution of responses (Methods). Figure 2A, B (yellow) show that time constant of the fir model is not significantly different than the data ($p$ value = 0.44, time constant for bias development). However, Fig. 2C (bottom row) shows that this model is not able to faithfully recapitulate the distribution of experimentally-observed responses and yields an energy landscape with a qualitatively different shape (Fig. 2C bottom, $R^2$ = 0.700, EMD = 0.072 and Fig. 2D bottom $R^2$ = 0.407, EMD = 0.117). Thus, a model that attempts to capture dynamics of human working memory across a number of delay periods as diffusion on a static energy landscape cannot simultaneously fit both the distribution of responses and the dynamics of development of biases.

These observations suggest that diffusion on a static landscape may not be sufficient to adequately capture the dynamics of working memory. This conclusion, however, critically relies on the fitting procedure. To provide additional model free confirmation of this conclusion, we examined the mutual information between the stimulus and the response.

Figure 3A shows the evolution of memory traces on a fixed energy landscape with two potential wells. Note, that early on during the delay period traces initiated from all three locations (red, green, blue) are readily distinguishable. However, as the system equilibrates and approaches steady state, the distinction between the red and green traces is lost. This information loss occurs because both the red and green initial conditions belong to the same basin of attraction – they eventually settle at the same energy minimum at steady state. The information loss is reflected in the characteristic deformation of the distribution of responses (Fig. 3B). Thus, at steady state, the only information about the stimulus that can be recovered from the response is the basin of attraction to which the stimulus belongs. All other information is lost. We therefore sought to determine whether the mutual information between the stimuli and the responses in working memory behaves in a manner consistent with the energy landscape model.

Experimental data (Figs. 1, 2) strongly suggest that the energy landscape has two wells separated by the energy maximum and that at the longer delay periods the system is approximately at steady state as evidenced by saturation of bias (Fig. 2A). The line segment is thus partitioned into two basins of attraction (to the left and right of the energy maximum). Because the information about different starting conditions for all points within the same basin of attraction is lost at steady state, the mutual information between the stimulus and the response at steady state is sensitive only to the overall number of energy wells. Specifically, for a uniform distribution of target locations (as was the case in experimental data) at steady state the mutual information between the stimulus and the response is equal to $-\sum_{i=1}^{2} p_i log(p_i)$, where $p_i$ is the probability of being in the $i$-th basin of attraction. Mutual information is maximized when for $p = 1/2$. Thus, the upper bound on the estimate of the mutual information between the stimulus and the response obtained at steady state from a two well potential system is log(2). Generally, for an $n$-potential well system the upper bound on information is log($n$).

Simulation of diffusion on a static landscape model with two energy wells, saturates close to this predicted value. In contrast, for the human behavioral data, we observe that mutual information saturates at a significantly higher value ($p$-value < 0.001). In order to explain this high amount of information in human data at steady state, the energy landscape would need to have about four wells. Indeed, the model fit to explicitly recapitulate the development of bias has three energy minima (Fig. 2C, bottom). This three well model, however, yields a qualitatively different distribution of responses from those observed experimentally. In other words, human working memory at steady state reflects higher information about the stimulus than would be expected from the static landscape model. This is the fundamental reason why static landscape model is unable to fit the totality of the data across all relevant time scales.

**Plastic landscape model faithfully recapitulates working memory dynamics.** How do we reconcile the saturating biases in working memory with higher information about the stimuli present at steady state? One important insight comes from observing that the static landscape model reliably fits the data after short delays but fails to predict long-term behavior. This observation suggests that the landscape changes on a slow timescale. Motivated by work on activity-dependent synaptic plasticity[33], we generalized the landscape model by allowing the memory trace to form a local deformation in the energy landscape (Fig. 4A). The rate of development of this activity-induced deformation as well as its spatial extent are added as parameters in the fitting procedure (Methods). Thus, the static model (Fig. 2) is a special case of the more general plastic landscape model, where plasticity parameters are set to zero. The fitting procedure for the plastic model was identical to that used for the static landscape model in Fig. 2. To get an intuitive understanding of how this local plasticity affects the dynamics, consider simulated trials in Fig. 4A. If the energy landscape remained static, with time both traces will decay to the energy minimum and become indistinguishable so long as both trials belong to the same basin of

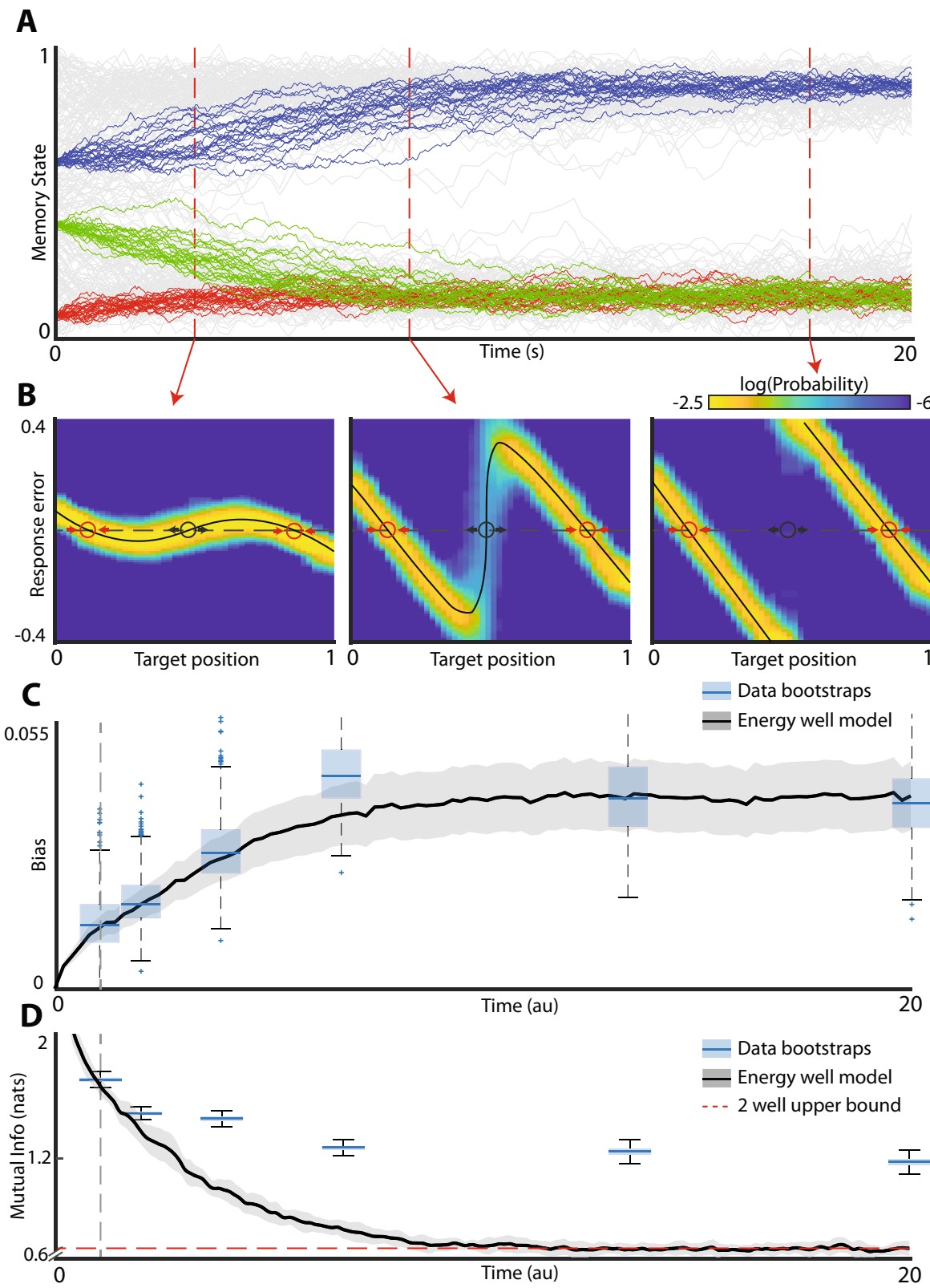

attraction. However, the deformation in the plastic model provides some local stability and impedes descent towards the energy minimum. As a result, trajectories starting from two locations within the same basin do not necessarily mix even at steady state.

The plastic landscape model (Methods) successfully recapitulates the dynamics of development of biases in working memory (Fig. 4B) and gets an unbiased estimate of the experimentally observed time constant for bias development (Fig. 4C, $p$ value = 0.22). Supplementary Fig. 6 shows that the estimation procedure for the plasticity parameters is robust (Supplementary Fig. 6). Given this result, we did not fit the plastic model to approximate the development of biases separately like we did with the static model. Furthermore, the model accurately predicts the distribution of responses both at short (Fig. 4D, left,

**Fig. 3 Energy landscape model at steady state contains less information about the stimulus than human working memory. A** Illustration of information loss in energy landscape model. Each color represents a unique target location. Drift and diffusion cause the red and green memory targets to become undistinguishable and all information about the initial target is lost as the system approaches steady state. **B** Response distributions of simulations in (**A**) at several different time points. Red circles denote stable fixed points and black circles denote unstable fixed points. **C** Bias time course for the experimental data and time-warped energy well system in (**A**). Blue boxplots are individual bootstrapped estimates of bias calculated from human participants. Thick black line and shaded area show mean and 95% confidence interval of the bias for the time-warped model. Note that after time warping, the energy well system is a qualitatively good match for the observed data. **D** Mutual information between the stimulus and response as a function of delay time. Blue boxplots are individual bootstrapped estimates of mutual information calculated from human participants. Thick black line show mean, shaded areas show 95% confidence interval of time-warped model. Red dashed line shows the theoretical upper bound of the mutual information that can be stored in a two well energy model.

$R^2 = 0.935$, EMD $= 0.040$) and the long (Fig. 4D right, $R^2 = 0.945$, EMD $= 0.025$) delays. Finally, unlike the static model in which the mutual information decays to ~log(2) at steady state, the plastic model retains significantly more information about the stimulus (Fig. 4E, $p$-value $< 0.001$). The decay in the mutual information closely resembles that seen in the experimental data. We confirmed that the plastic model is a better approximation to the experimental data than a discrete attractor (static) model using Akaike Information Criterion (AIC) (static model AIC $= 1.61$e5, plastic model AIC $= 1.92$e5, $p$ value $< 0.001$) and Bayesian Information Criterion (BIC) (static model BIC 1.61e5, plastic model BIC $= 1.92$e5, $p$ value $< 0.001$).

There are two essential components to the plastic model. The first is the static component of the landscape. The second component is the plasticity of this landscape during the delay period. Consistent with the distribution of the data at short delays and with previous observations arguing for the discrete rather than continuous attractors, the static component of the landscape does indeed reveal two discrete energy wells (Supplementary Fig. 6). Thus, in total, our results imply that plasticity superimposed onto this discrete attractor landscape is required to fully capture dynamics of working memory.

**Plastic model explains proactive interference observed in experimental data.** One consequence of the plasticity of the energy landscape is that different trials of the working memory task can become interdependent. This interdependence arises if the plasticity imposed by the previous trial does not relax completely by the time the next trial is initiated. Indeed, proactive interference (PI) between trials is well-known to corrupt working memories[34,35]. PI was also readily apparent in our experimental data (Fig. 5A). Simulations of the plastic model revealed that it can indeed give rise to PI and that this PI is similar to that observed experimentally (Fig. 5B). Consistent with previous work on this subject the overall shape of the PI is similar to the derivative of Gaussian (DoG)[36]. One minor deviation between the experimental observations and model predictions is that DoG is slightly shifted—yields a small non-zero interference between two trials in which target stimuli were presented at the same location.

The model predicts that the magnitude of the PI should grow as a function of delay time (Fig. 5D). This prediction was also confirmed with experimental observations (Fig. 5C). The details of PI demonstrated in this task are qualitatively similar to those observed with visual working memory task in primates[37]. Thus, we show how both the development of biases in human visual working memory and the inter-trial interference can both be explained with the plasticity of the energy landscape. Given that the model fit was completely agnostic to the existence of PI, the qualitative agreement between model predictions and experimental observations is rather good.

The local deformation of the energy landscape in the plasticity model naturally stabilizes the memory trace against the corruptive influence of diffusion. However, in contrast to discrete attractors, plasticity does not increase the bias. Instead, the tradeoff is between the noise-stabilizing properties of plasticity and the level of PI. This raises an interesting question: Do discrete attractor offer additional noise stabilization benefits over continuous attractor systems once plasticity is introduced?

To begin to address this question, we first compared the performance of discrete and continuous attractor systems without plasticity on the accuracy of working memory performance. In the noise free case, continuous attractors yield perfect performance, as expected (Fig. 5E, black line). In contrast, discrete attractor system produces biased responses even in the noiseless limit. Thus, the fidelity of memory storage is degraded in the discrete attractor system in the noise free regime (Fig. 5E, red line). Consistent with previous work, in a noisy regime, the discrete attractor model yields better performance at longer delay times, $p$ value $< 0.001$ (Fig. 5E, blue and green lines, $p$ value based on difference at the longest delay). Thus, as shown previously[10,18,26], discrete attractors are superior to continuous attractors for memory storage in noisy systems because discrete energy wells stabilize the memory trace against the corrupting influence of noise.

It is not clear, however, whether noise stabilization afforded by discrete attractor dynamics is also beneficial in systems with plasticity. To address this question, we simulate the performance of both continuous and discrete attractor systems with varying amounts of plasticity. We specifically focused here on the effect of the overall plasticity strength ($\beta_p$)—a time constant that sets how quickly the local deformation of the energy landscape develops. Other parameters such as noise and ($\sigma$), and shape of the plastic deformation ($\sigma_p$) (Eqs. 3–5, Methods) were kept constant at the values that best approximated the experimental results. Our experimental design did not allow for robust estimation of the rate of dissipation of the local energy deformation between trials ($\lambda_T$, Eq. (7), Methods). Thus, we kept this parameter constant at 0.8—a value that yielded the best qualitative fit to the amount of PI between trials observed experimentally. So long as $\lambda_T$ is long enough such that plasticity does not dissipate completely between trials, PI will ensue. Note that $\lambda_T$ sets the magnitude of interference between trials rather than the shape of the interference (Fig. 5B). The models were simulated on sequences of trials to assure that the penalty for PI is considered in the estimates of the overall accuracy. The relative cost of the PI, generally depends on the relative timescales of $\lambda_T$ and the intertrial interval. Thus, the following theoretical results do not undergo qualitative changes with different intertrial decay values. We find that if a non-zero amount of plasticity is added to the model the continuous attractor system with plasticity always outperforms the discrete attractor system with plasticity ($p$-value $< 0.001$, Fig. 5F). This is surprising as it suggests that there is no intrinsic noise stabilization benefit of discrete attractor systems so long as the stationarity assumption is relaxed and plasticity is allowed to deform the energy landscape and thus stabilize the memory trace against noise.

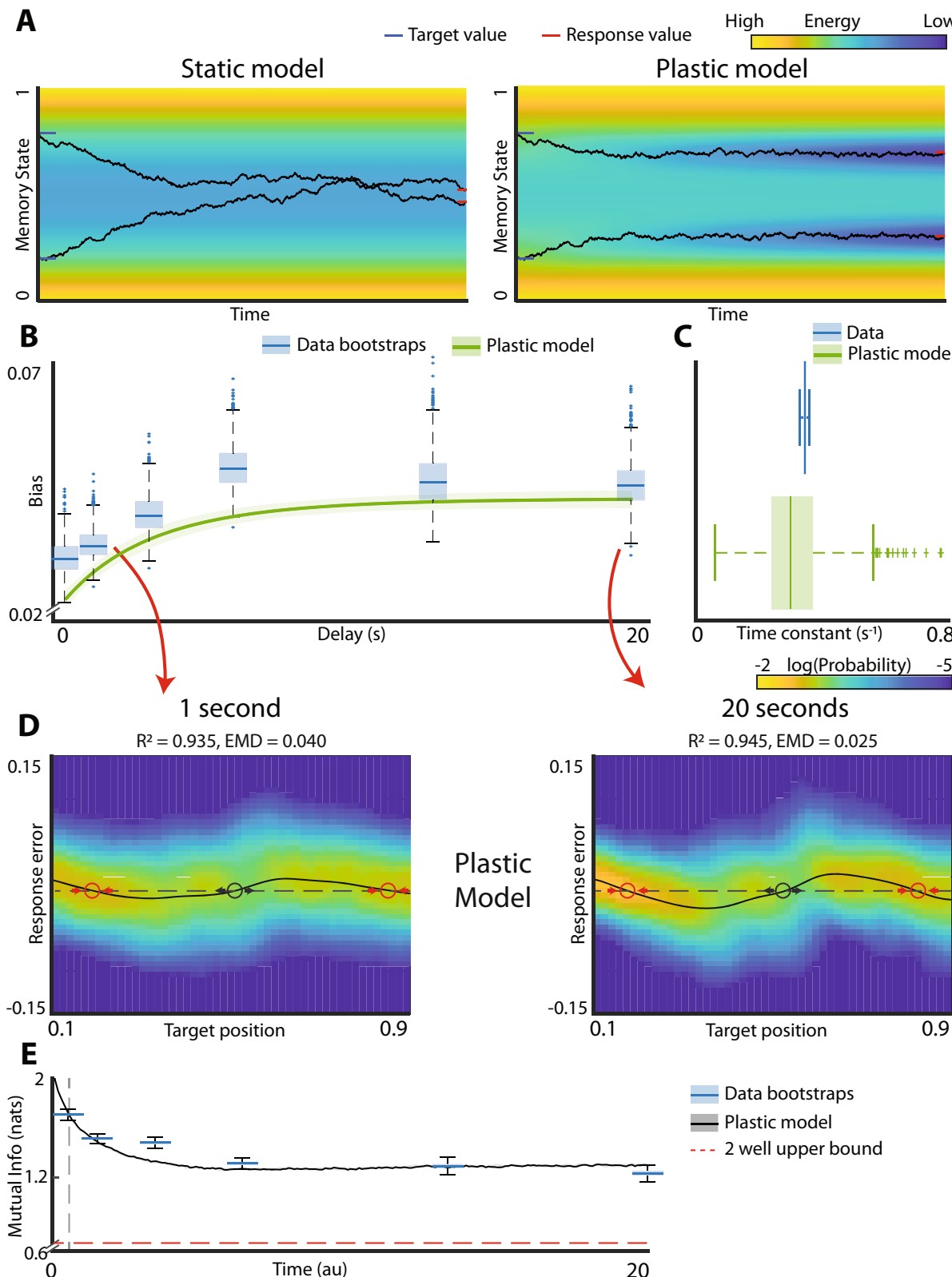

## Discussion

Here, we attempted to approximate the dynamics of working memory in humans on a simple visual task across different delay periods using diffusion on a fixed energy landscape. Diffusion on an energy landscape is a general model of the dynamics of attractor neural networks that can store stable memory representations in the absence of stimulus. We find that while static energy landscape models are sufficient to accurately fit the response distribution at a single delay timepoint, they cannot fit the performance across different delays. This failure occurs because even after memory biases saturate, the responses contain more information about the stimuli than would be expected from a static landscape model. The addition of a plasticity term expressed as local deformation of the energy landscape formed

**Fig. 4 Addition of plasticity to the energy landscape model fits the totality of data across all time scales. A** Illustration of the effect of plasticity on memory traces. Toy system with a single energy well (energy is represented by color). Black curves show memory traces starting on opposite sides of the energy minimum. Without plasticity (left), the traces coalesce together, and information is lost. With plasticity (right), the system adds a small negative deflection to the landscape at each step which lets it "burrow" into the landscape and resist drift. Thus, the two memory traces do not necessarily converge to the same stable state. **B** Bias time course for plastic model fit to experimental data. Blue box plots are individual bootstrapped estimates of bias calculated from observed data. Green line and shaded area are the bias mean and 95% confidence interval for the plastic model fit to the data. **C** Distribution of time constants for the data and the plastic model. **D** Response distributions of plastic model at 1 and 20 s. Red circles are stable fixed points, black circles are unstable fixed points. Reconstruction $R^2$ values and earth mover's distance (EMD) for comparison to experimental data (shown in Fig. 2C, D top) are reported for model fits. **E** Mutual information time course for experimental data and plastic model. Blue boxplots are individual bootstrapped estimates of human data. Thick black line and shaded area show mean and 95% confidence interval of the model. Start time of plastic model is adjusted to align with experimental data (gray dashed line). Red dashed line shows the mutual information of the 2-well model at steady state.

during the delay period fits the data across all delay periods, explains why the responses contain information about the stimulus even at steady state, and accurately predicts inter-trial interactions. Thus, our results suggest that stabilized transients contribute to working memory dynamics. The overall dynamics of the memory state, therefore, are influenced both by the pre-formed attractors that shape the trajectory of the memory state, and the plasticity mechanisms that stabilize the trajectory and impede its decent into the attractor.

Attractor neural networks offer an appealing framework which can explain how the memory trace can be transiently stored after the stimulus is withdrawn and resist noise-driven perturbations[18]. These models received considerable experimental and theoretical support[10]. While continuous attractors are in principle best suited for storing memories of continuous variables such as spatial locations, noise fundamentally limits their storage capacity[26]. In contrast, discrete attractors have been hypothesized to produce a mechanism for neural systems to counteract noise and stabilize stored memory[18]. This stabilization comes at the cost of introducing memory biases[10]. Our experimental results and many other similar findings[8–12] reveal biases in memory storage and are thus, at least superficially, consistent with the discrete attractor model.

There is, however, an important departure of our experimental findings from the predictions of the discrete attractor framework. Once a discrete attractor network settles into a steady state and biases saturate, the only information remaining about the original memory representation is which attractor it was closest to at the onset. Our results show that human visual working memory contains more information about the original stimulus even after biases saturate. Thus, our experimental findings depart from the predictions made by both continuous and discrete attractor networks.

Continuous attractor networks have been successfully deployed to model performance on visual working memory[19,38,39] and other settings such as path integration[40,41]. To create a network with a continuum of attractor states, biologically unrealistic symmetry is required[25]. In more biologically realistic models in which properties of individual neurons and synapses are allowed to vary[42], a continuum of attractor states quickly disintegrates, and a few discrete attractors form[43]. This observation has been used to argue that, while continuous attractor networks, are a natural choice for storing memories of continuous variables such as spatial locations, they are not biologically plausible. However, synaptic strength evolves dynamically as a function of neuronal activity[43,44]. Remarkably, introduction of synaptic scaling into more biologically plausible heterogenous neuronal networks was able to effectively homogenize the network and restore approximately continuous attractor dynamics providing a strong argument for biological plausibility of continuous attractor dynamics[29]. The interactions between the heterogeneity of network architecture and activity-dependent synaptic plasticity

prompted several theoretical investigations of the effect of plasticity on dynamics of memory storage in ring attractor networks[33,44]. Kilpatrick[44] demonstrates that when the environment is changing rapidly, the best prediction strategy for the next stimulus is most heavily weighed by the location of the immediately preceding stimulus. This insight naturally leads to the kinds of intertrial PI that has been observed by us and by others[36,37]. Such intertrial interactions arise naturally in ring attractor networks with activity-dependent plasticity[44]. Here, we chose a general modeling framework agnostic to the specifics of network architecture. This phenomenological model is consistent with mean field approximations of attractor neural networks with synaptic plasticity[33]. Our results are consistent with the proposals that working memory storage is not mediated solely by pre-formed attractor dynamics but also involves activity-dependent plasticity. Simulations of the plastic landscape model show that local landscape deformations impede the drift down the energy gradient. This has the benefit of increasing the information stored about the stimulus at long delay intervals but causes proactive inference between trials.

The overall shape of the interference between two consecutive trials as observed experimentally herein and in other related work[36,37] is similar to a DoG. This shape emerges naturally if one assumes that, as the memory trace traverses the energy landscape on the previous trial, it produces a Gaussian depression in the landscape. If this Gaussian depression does not completely relax in the inter-trial period, then the subsequent responses should be attracted towards the previously recalled location so long as the stimulus on the next trial is near the one on the preceding one. More complex experimental paradigms that involve interactions between multiple stimuli revealed both attractive and repulsive intertrial interactions[45,46]. Fritsche et al.[47] also observed both repulsive and attractive interference between trials. In this case the direction of the interference depended on the timescale—over short time scales attractive interference is observed; while on longer time scales repulsive interactions appear to be present. In principle, the plastic landscape model can yield both attractive and repulsive interactions depending on the choice of the plasticity kernel. An attractive component is required in order to impede the descent of the memory trace into the energy well and preserve the mutual information between the stimulus and the response, but a more exotic plasticity kernel can in addition include a repulsive interaction. Our experiment was not designed to specifically probe the finer details of the shape of the plasticity kernel or the timescale on which the depression imposed by the previous trial dissipates in the inter-trial period. The modeling framework proposed herein can be further refined in future experiments specifically designed to address these important issues.

PI of a similar kind to that observed herein has been studied using continuous attractor models in macaques[29,31]. While continuous attractor models with plasticity capture the effects of PI,

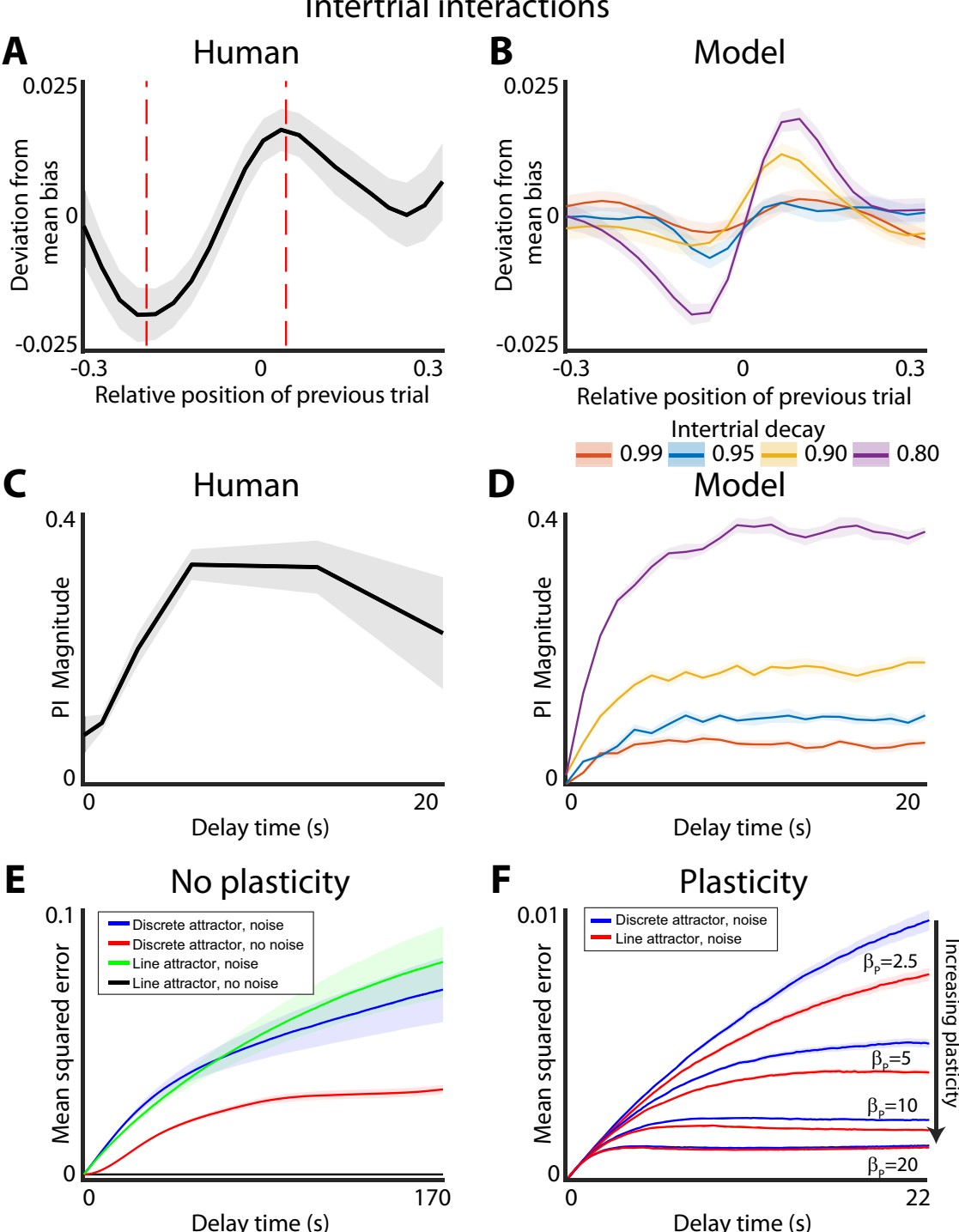

**Fig. 5 Plastic model produces proactive interference and adds robustness to noise. A** Proactive interference of experimental data. Effect of previous trial on bias of current trial based on the relative position of the previous trial. Line and shaded area show mean and 95% confidence interval for the deviation of mean bias. Red dashed line shows the points where the deviation due to plasticity is at its minimum and maximum. **B** Same as (**A**) but for the plastic model simulated with different intertrial decay values. **C** Time course of proactive interference. Spatial scale of bias is calculated by taking the maximum minus the minimum deviations from (**A**) (points denoted by red dashed lines). Note that the effect of the plasticity saturates after 6 s. **D** Same as (**C**) but for the plastic model simulated with different intertrial decay values. **E** Effect of noise on different attractor systems. Line and shaded area show mean and variance of the mean squared error of the system over time. Simulations assume a uniform starting distribution and a two well system as in Fig. 3. **F** Effect of plasticity on discrete attractor system. Line and shaded area show mean and variance of the mean squared error of the system over time. Different levels of plasticity are plotted, but regardless of plasticity level the non-plastic model always has more overall error than the plastic model. Note that the fit model has a plasticity weight, $\beta_P$, of ~10.

continuous attractors give rise to unbiased memories. Other work using continuous attractors has suggested that the combination of plasticity in the form of short-term facilitation and depression over many repeated trials can give rise to sustained biases in memory representations[33]. This work is consistent with theories that long-term synaptic plasticity is a mechanism for learning[44]. Thus, our model of plasticity on a static energy landscape can be understood as plasticity on two separate time scales. The fast time scale plasticity increases the fidelity of memory storage at the cost of PI. However, over hundreds of trials this plasticity may build on a slow time scale to shape the system's systematic biases as expressed by the energy landscape. This intuition is consistent with previous work reporting PI over time scales from single trials[48] to several days[49]. While PI degrades performance on our working memory task where the location of target stimulus was sampled randomly from a uniform distribution, in more etho- logically salient tasks increased autocorrelation in memory may prove to be a useful feature[35,44,50,51]. Eissa and Kilpatrick[52] recently showed that networks with long-term synaptic plasticity can, overtime yield energy landscapes that conform to the dis- tribution of stimuli in the environment that efficiently store information in ethologically salient settings.

Interestingly, we find that in systems with plasticity, discrete attractors no longer increase the system's overall robustness in the presence of noise. Yet, our results and those of Panichello et al.[10] strongly argue that discrete attractors are involved in mediating visual working memory. This raises the question: Why have dis- crete attractors at all? One possibility is that the discretization afforded by discrete attractor dynamics can be used to compress the memory representations into manageable "chunks"[53]. By categor- izing memories into compressed chunks, the system can store much more behaviorally salient information than if it stored the specifics about each memory[54]. Thus, discrete attractors may not have emerged in working memory to confer robustness to noise but to effectively store complex memories in manageable chunks.

**Limitations**. There are several limitations of our study that should be addressed in the future work. We aggregated the data across many individuals. While this choice is justified by the fact that the distribution of responses was approximately conserved across participants, future work should determine if the plasticity model applies to each participant individually. This analysis may reveal interesting inter-individual differences in the relative contributions of the pre-formed attractor system and plasticity. The second notable limitation of our study was that it was not explicitly designed to quantify the dynamics of dissipation of plasticity during inter-trial interval. If it is indeed true that the plasticity in working memory confers a benefit in ethologically salient tasks with notable temporal correlations, the time scale of plasticity ought to reflect the time scale of the temporal correla- tions in the environment.

## Data availability
Data is available at https://doi.org/10.17605/OSF.IO/FXWMR[55]. No restrictions on sharing. All data are packaged into ParticipantData.rar. Each participant has a separate folder with one JSON file for each 10-block set of trials they completed.

## Code availability
The code used to produce the results in this Manuscript can be found at https://doi.org/10.17605/OSF.IO/FXWMR[55].

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

## Acknowledgements

We would like to thank Josh Gold for helpful discussion on the design and implementation of the working memory task. We would also like to acknowledge the help of Adeeti Aggarwal and Rui Pei performing the initial versions of the task and helping us fine tune the parameters of stimulus presentation. This work was funded by grant R01NS113366, awarded to A.P. from the National Institute of Neurological Disorders and Stroke. Along with funding from the Google PhD Fellowship Program awarded to C.B. The funders had no role in study design, data collection and analysis, decision to publish or preparation of the manuscript.

## Author contributions

Both authors contributed equally to this work. C.B. wrote code, performed experiments, and analyzed the data. A.P. helped design experiments and analyses.

## Competing interests

The authors declare no competing interests.
