## [Peer Review File · Communications Psychology]

27th Apr 23

Dear Dr Brennan,

Thank you for your patience during the peer-review process. Your work "Plasticity and attractor dynamics in human visual working memory." has now been seen by 3 reviewers, and I include their comments at the end of this message. As a computational model forms part of the main contribution of the paper, your submission required review of the model code in addition to the evaluation by our external referees and the standard editorial evaluation. We were facing difficulties soliciting a report on the code from a reviewer dedicated solely to the task in a timely manner; in agreement with the Chief Editor, I therefore undertook testing of your custom code myself.

The reviewers find your work of interest, but raised some important points. We are interested in the possibility of publishing your study in *Communications Psychology*, but would like to consider your responses to these concerns and assess a revised manuscript before we make a final decision on publication.

We therefore invite you to revise and resubmit your manuscript, along with a point-by-point response to the reviewers. Please highlight all changes in the manuscript text file.

Editorially, we consider that there are two major aspects that are consistently raised by the reviewers that will need to be carefully addressed. First, the revision will need to include the supplementary analysis requested by reviewer #3 addressing the robustness of the fitting procedure. Second, the introduction will need to be revised to provide a clearer message and a clearer ratio on the logic of the model and its epistemic gain (Reviewer #1 and Reviewer #2 have many suggestions on how to implement this).

As you revise the manuscript, please address two issues regarding the code listed below the referees' reports as well as the other editorial policy formatting requests listed below and in the linked checklist (below). Please pay particularly close attention to our guidelines for statistics reporting and interpretation (also available at <https://www.nature.com/commspsychol/submit/submission-guidelines#statistical-guidelines>), and the guidelines regarding data presentation in the Figures.

Please use the following link to submit your revised manuscript, point-by-point response to the referees' comments (which should be in a separate document to any cover letter) and the completed checklist:

[link redacted]

We hope to receive your revised paper within 8 weeks; please let us know if you aren't able to

submit it within this time so that we can discuss how best to proceed. If we don't hear from you, and the revision process takes significantly longer, we may close your file. In this event, we will still be happy to reconsider your paper at a later date, provided it still presents a significant contribution to the literature at that stage.

Please do not hesitate to contact me if you have any questions or would like to discuss these revisions further. We look forward to seeing the revised manuscript and thank you for the opportunity to review your work.

Best regards,

Eva R. Pool

Eva R. Pool, PhD
Editorial Board Member
Communications Psychology
orcid.org/0000-0001-5929-1007

EDITORIAL POLICIES AND FORMATTING

Editorial Policy: [Policy requirements](https://www.nature.com/documents/nr-editorial-policy-checklist.pdf) (Download the link to your computer as a PDF.)

Furthermore, please align your manuscript with our format requirements, which are summarized on the following checklist:

[Communications Psychology formatting checklist](https://www.nature.com/documents/commsj-style-formatting-checklist-review-perspective.pdf)

and also in our style and formatting guide [Communications Psychology formatting guide](https://www.nature.com/documents/commspsychol-style-formatting-guide-accept.pdf) .

* **CODE AVAILABILITY:** All Communications Psychology manuscripts must include a section titled "Code Availability" at the end of the methods section. In the event of publication, we require that the custom analysis code supporting your conclusions is made available in a publicly accessible repository; at publication, we ask you to choose a repository that provides a DOI for the code; the link to the repository and the DOI will need to be included in the Code Availability statement. Publication as Supplementary Information will not suffice. We ask you to prepare code at this stage, to avoid delays later on in the process.

* **DATA AVAILABILITY:**

All Communications Psychology manuscripts must include a section titled "Data Availability" at the end of the Methods section or main text (if no Methods). More information on this policy, is available at <http://www.nature.com/authors/policies/data/data-availability-statements-data-citations.pdf>.

At a minimum the Data availability statement must explain how the data can be obtained and whether there are any restrictions on data sharing. Communications Psychology strongly endorses open sharing of data. If you do make your data openly available, please include in the statement:

We recommend submitting the data to discipline-specific, community-recognized repositories, where possible and a list of recommended repositories is provided at <http://www.nature.com/sdata/policies/repositories>.

If a community resource is unavailable, data can be submitted to generalist repositories such as [figshare](https://figshare.com/) or [Dryad Digital Repository](http://datadryad.org/). Please provide a unique identifier for the data (for example a DOI or a permanent URL) in the data availability statement, if possible. If the repository does not provide identifiers, we encourage authors to supply the search terms that will return the data. For data that have been obtained from publicly available sources, please provide a URL and the specific data product name in the data availability statement. Data with a DOI should be further cited in the methods reference section.

REVIEWERS' EXPERTISE:

Reviewer #1 computational models of working memory
Reviewer #2 computational models of visual working memory
Reviewer #3 continuous attractor models of working memory

REVIEWERS' COMMENTS:

Reviewer #1 (Remarks to the Author):

Summary

The authors aimed to test the conditions in which discrete and continuous attractor networks fail to capture working memory storage in humans. To this end, the authors specifically tested the effect of delay time on working memory storage dynamics in these models and show that neither model is sufficient to capture human working memory. They propose that activity-based plasticity is a critical element to add to attractor models. By comparing human data and model outputs, the authors show how the plasticity model better fits the human data at short and long time delays than the discrete attractor model. The plasticity model is interesting because of synaptic changes observed during learning and working memory tasks. While the idea is compelling, there are some major concerns that need to be addressed before moving forward. Some of the biggest concerns include the clarity of the paper and the lack of detail provided on the plasticity model. The other major and minor concerns are voiced below.

Major Concerns

1. The ms could benefit if the narrative is better organized and the aims are more clearly articulated. For example, only in the general discussion the authors provide a form of motivation for testing delay and plasticity effects on attractor neural network. This should have been properly explained in the introduction instead of summarizing the results at the end of the introduction. If the aim of the paper is to show that plasticity is critical for modeling human WM, then the concept of plasticity should be much more elaborated in the introduction.

- a. The term memory bias is introduced without explanation in the introduction. The authors might consider placing the explanation for Figure 1B and 1C in the introduction instead of the results.
- b. Motivate in the introduction the use of different attractor models and the conditions where each of them fails to set the stage to the plasticity idea before testing it.
- c. In most of the introduction and first few pages, there are references to the fact that the plasticity model is better at fitting human data, but the reasons for this are kept elusive until much later. It would be prudent to simply state that the plasticity model does a better job of fitting human data at 1) longer delays 2) protecting against noise and 3) intertrial relationships.
- d. A lot of relevant and needed context from literature review is provided in the general discussion. Some of this should be moved to the introduction. For example the authors should clearly state the predictions made by the discrete and continuous attractor models in the introduction. It should also be made explicit what the authors are aiming to test by using the plasticity model. (e.g., lines 370-373)

2. The parameters of the plasticity model, specifically, the amplitude and width, seem critical to defining how the plasticity model will perform. It would be beneficial to have a discussion about how those parameters were selected and in what parameter ranges the qualitative results that are shown hold. Furthermore, the plasticity model includes the transient wells - the emergence and dissolution of those wells should be further discussed.

3. The usefulness and meaning of the results reported in the last paragraph (lines 325-337) is unclear. It is confusing to report that discrete attractor models are better than the continuous one (but that the plasticity model is even better than the discrete model when delay time is considered) and then right below it report a seemingly contradicting result where the continuous model outperforms the plasticity model. The authors are encouraged to motivate this analysis better in the context of the study aims and reconcile the interpretation of this result in respect to the other results reported in the study.

4. From the paper: "We find that regardless of the amount of plasticity added to the model the continuous attractor always outperforms the discrete attractor (p -value < 0.001)." This is a bit confusing, because in theory it would make sense that as the amount of plasticity approaches 0, the "plasticity" model should approximate the discrete attractor model. If it does not, does this suggest there were other changes added in the plasticity model (on top of just plasticity) that is not in discrete attractor model that may account for the results. Testing this (as plasticity goes to 0, the plasticity and discrete model have the same results) would be an important control case to make sure that all else is equal between the models.

Minor Concerns

1. From the paper: "with the addition of noise the discrete attractor model beats out the continuous attractor model given sufficient time, p -value < 0.001 (Figure 5E, blue and green lines)". Is this p -value calculated using all time delays, or as the text suggests, only uses the longer time delay values (is the p -value calculated on the whole line or part of the green vs. blue line).

2. The authors report that 0.8 is the best intertrial values and since 0.8 is also the lowest value tested, there should be some values less than 0.8 that are tested. Some statistics should be performed on the curves to show which value is the best/why it was chosen.

3. In Figure 5D, "Spatial scale of bias" is a confusing y-axis title. The figure legend is helpful in explaining that it was the difference between the max and min deviations (the red lines in A). Perhaps consider changing the axis title to "PI magnitude" (as referred to in the text) or to a name that more intuitively described the results reported there.

4. There is no reference in the text for Figure 5F.

Reviewer #2 (Remarks to the Author):

Brennan and Proekt used a computational modeling approach to test the attractor dynamics in working memory. They found that the discrete attractor model by itself could not fully explain human behavior in a spatial working memory task. When plasticity was introduced to the discrete attractor model, the model better accounted for working memory bias across different time points as well as inter-trial serial dependence. They conclude that maintenance of working memory content relies on a stable structure with a plasticity mechanism.

Overall, I thought that this was a well-motivated paper that addresses the temporal dynamics of working memory. I found the paper enjoyable to read, the methods appropriate, and the results carrying implications for the literature on the discrete attractor model. My comments are fairly minor in nature. For example, there is literature on serial dependence bias that could be related to the intertrial dynamics observed in the current study and speak to the nature of the bias. This and other points are detailed below.

Main:

1. Mechanisms for working memory bias.

a. There may be multiple sources for working memory bias. Behavioral observations show that the direction of bias may be attractive or repulsive, which has been linked to different mechanisms (e.g., Chunharas et al., 2022; Czoschke et al., 2019). For example, Fritsche et al. (2017) have suggested that positive bias reflects attraction from the previous decision, whereas negative bias reflects neural adaptation that repels current perception away from previous stimuli. I am wondering whether/how the model in the current paper may address these biases that happen at different levels (low-level perceptual vs. high-level decisional).

b. The authors assumed that the attractive bias in human data is a result of plasticity. However, related to my previous point, this type of attractive bias may be more likely to reflect a decisional bias rather than a perceptual one. Is it possible that despite the apparent similarity between the human data and the model simulation, the two may be reflecting different mechanisms?

c. Regarding the intertrial interactions, this type of proactive interference (also referred to as 'serial dependence' in other contexts) may last more than one trial and the bias has been reported to change its direction from attractive to repulsive on a longer time scale. For example, Fritsche et al. (2020) observed that the attractive bias has a long-lasting effect across a couple of trials, and interestingly the bias turned to be repulsive after 5 trials. Did the authors observe similar cross-trial dynamics in the current dataset? If so, could this pattern be accounted for by the plasticity model?

2. Although Figure 4 showed that the energy landscape model with added plasticity could nicely match the temporal dynamics of bias from human data, I am not certain whether the model also tracks the change in precision across time. Specifically, comparing 4D (1-second delay) with 4E (20-second delay), memory precision seems to be quite similar. In contrast, human data in Figure 2C and 2D showed a clearly higher precision at 1 second than at 20 seconds. It was not entirely clear to me how the precision was accounted for by the model. More details on this could be helpful.

3. Could the authors clarify why plasticity is accomplished through the local deformation landscape? How many deformation points could there be and is it always going to match the number of discrete attractors? Lastly, why would a depression predict an attractive inter-trial bias, rather than an adaptation-like effect?

Minor:

1. It was mentioned a few times that "... human working memory at steady state reflects significantly higher information about the stimulus than would be expected from the static landscape model" [lines 238-240]. What exactly is the information that is not accounted for by the static landscape model?

2. The simulation results in Figure 5B showed a symmetrical pattern where the mean bias was around 0 when the inter-trial distance was 0. However, the sinusoidal pattern from the human

subjects showed a shift towards the left, such that the bias was positive at distance of 0. Why would one expect to see a pattern like this?

3. In Figure 5E: the time scale ranges from 0 to 170 s for the No plasticity model, but 0-22 s for the Plasticity model. Is there any reason for this?

4. It may be easier for the readers to follow if the terminologies could be consistent throughout. For example, is the “energy landscape model” the same as the “energy well model”? and also the “line attractor” and “continuous attractor”?

Reference:

Chunharas, C., Rademaker, R. L., Brady, T. F., & Serences, J. T. (2022). An adaptive perspective on visual working memory distortions. *Journal of Experimental Psychology: General*.

Czoschke, S., Fischer, C., Beitner, J., Kaiser, J., & Bledowski, C. (2019). Two types of serial dependence in visual working memory. *British Journal of Psychology*, 110(2), 256-267.

Fritsche, M., Mostert, P., & de Lange, F. P. (2017). Opposite effects of recent history on perception and decision. *Current Biology*, 27(4), 590–595, <https://doi.org/10.1016/J.CUB.2017.01.006>.

Fritsche, M., Spaak, E., & De Lange, F. P. (2020). A Bayesian and efficient observer model explains concurrent attractive and repulsive history biases in visual perception. *Elife*, 9, e55389.

Reviewer #3 (Remarks to the Author):

Biases in working memory tend to accumulate over time. This has led to proposals that working memory may be well-described as a dynamical system in which memories drift towards attractor states. Here, the authors propose that the energy landscape underlying this dynamical is itself dynamic, changing over the course of the trial due to short-term synaptic plasticity. They argue that this plastic model parsimoniously captures key features of both intra- and inter-trial phenomena.

Overall, the paper is conceptually interesting and the use of a single modeling scheme to address both single-trial dynamics and intertrial interference is innovative. However, I have some concerns with the interpretation of the results and I feel the manuscript could be improved with further analysis.

General Comments

1. It would be helpful to have a sense of how participants performed overall before diving into the details of the model and derived statistics. For example, what was the error distribution like at each delay, and how variable was mean error across subjects? Is it possible to estimate the lapse rate (‘guessing’) in this linear space?

2. The static model analyzed by the authors differs from previous efforts in two major ways. First, the model is fit to aggregate features of the data (error distributions or bias) using Monte Carlo methods, rather than fitting the data to single trials responses using a non-stochastic estimate of the response PDF. This has the potential to influence both the accuracy and precision of parameter estimation, and therefore the behavior of the best-fitting model. This concern could be addressed by demonstrating that the fitting procedure can adequately recover a range of parameters used to

generate simulated data.

3. Second, the authors account for baseline bias and variance in behavior (at a hypothetical zero-delay) by allowing the model to evolve for a fit number of baseline timesteps. However, previous work suggests that the values of drift and noise during such baseline periods differs substantially from that during memory, and accounting for this in the model results in superior performance. Therefore, this has the potential to substantially affect the model fits presented in Figure 2. This concern could be addressed by fitting separate values of drift and noise during baseline.

4. Two different fitting procedures were applied to the static model, while one fit of the plastic model is shown in Figure 4. However, I couldn't find which method was applied here. It would be helpful to clarify this explicitly in Figure 4 or the related text.

5. In the methods section it wasn't clear how exactly the AIC values for model comparison were computed – could the authors clarify? Additionally, since AIC is known to favor more complex models, it would be helpful to report the BIC values alongside these for comparison. Or, perhaps preferably, reporting cross-validated fits instead of AIC/BIC given the impressive size of the data set.

6. The authors make the strong claim in the abstract that 'dynamics of human working memory violate predictions made by the discrete attractor theory', but ultimately conclude that 'in total, our results thus far agree with the finding that discrete, rather than continuous attractors likely contribute to the storage of working memory, but show that plasticity is required to fully capture the dynamics'. Clearly, these are very different statements, and the latter is much more consistent with the results as they currently stand.

7. These results seems potentially complimentary to (though quite distinct from) those of Eissa and Kilpatrick (biorxiv, 2022). A brief consideration of this in the discussion might be interesting.

Specific Comments

1. There appears to be a typo on line 266 – “[The] plastic landscape”?

Editorial board member Eva Pool:

For reproducibility purposes we verified that the code of the model is functional and produces the described results.

The code runs, it is described in sufficient details and reproduces the main results and figures reported in the paper. Only two changes are necessary (1) please change the input 'color' to something else like for instance 'colorPlot' in the auxiliary function plotShadedCI.m to avoid conflicts with built in functions. (2) Please add the parfor_progressbar.m function in the Library folder.

We would like to thank the reviewers for their insightful critiques of our manuscript. Guided by these critiques, we significantly modified the manuscript and incorporated additional tests of the models requested by the reviewers. Specifically, we verified that our Monte Carlo method yields a good estimation of the parameters using simulations. We also added tests that confirm that our fitting of the plasticity parameters is robust. Finally, we re-structured the manuscript to reflect the suggestions of the reviewers. We believe that as a result, our manuscript is much improved. Thank you.

Below we address each of the critiques in detail.

Reviewer 1:

1. The ms could benefit if the narrative is better organized and the aims are more clearly articulated. For example, only in the general discussion the authors provide a form of motivation for testing delay and plasticity effects on attractor neural network. This should have been properly explained in the introduction instead of summarizing the results at the end of the introduction. If the aim of the paper is to show that plasticity is critical for modeling human WM, then the concept of plasticity should be much more elaborated in the introduction.
 - A. The term memory bias is introduced without explanation in the introduction. The authors might consider placing the explanation for Figure 1B and 1C in the introduction instead of the results.
 - b. Motivate in the introduction the use of different attractor models and the conditions where each of them fails to set the stage to the plasticity idea before testing it.
 - c. In most of the introduction and first few pages, there are references to the fact that the plasticity model is better at fitting human data, but the reasons for this are kept elusive until much later. It would be prudent to simply state that the plasticity model does a better job of fitting human data at 1) longer delays 2) protecting against noise and 3) intertrial relationships.
 - d. A lot of relevant and needed context from literature review is provided in the general discussion. Some of this should be moved to the introduction. For example the authors should clearly state the predictions made by the discrete and continuous attractor models in the introduction. It should also be made explicit what the authors are aiming to test by using the plasticity model. (e.g., lines 370-373)

Our response: Thank you. To address this critique, we added a paragraph to the introduction which clearly lays out the predictions made by both the continuous and the discrete attractor models. Specifically, we explain that the dynamics of both models can be approximated as drift and diffusion on a fixed energy landscape. By the Boltzmann equation, the distribution of the responses observed at steady state, therefore, is determined by the shape of the landscape. Thus, in the limit of long delays, the mutual information between the stimulus and the response for the discrete attractor model is determined by the number of discrete energy wells. This prediction is not consistent with experimental observations at longer time delays. In this paragraph we also explain in more detail the concept of bias. We believe that this more clearly lays out the predictions of the existing models and sets the stage for the plasticity model introduced in our manuscript.

2. The parameters of the plasticity model, specifically, the amplitude and width, seem critical to defining how the plasticity model will perform. It would be beneficial to have a discussion about how those parameters were selected and in what parameter ranges the qualitative results that are shown hold. Furthermore, the plasticity model includes the transient wells - the emergence and dissolution of those wells should be further discussed.

Our response: *We apologize for not spelling this out as clearly as we had hoped in the original submission. To address this concern, we have re-written the methods section to more clearly indicate that the plasticity parameters (β_p and σ_p) were fit to the data using Monte Carlo methods along with the parameters that describe the shape of the landscape and the diffusion. In order to explore the possible inter-trial interference effects that can arise in the plastic landscape model we additionally introduced a parameter lambda which governs how the deformation at the end of the previous trial relaxes between trials. This parameter does not affect the shape of the inter-trial interactions as those are determined by the shape of the Gaussian deformation but only the strength of these inter-trial effects. Because the trials in our experiment were self-paced it was not possible to infer this decay time constant reliably from the experimental data. This should be addressed in future work in which inter-trial interval should be varied parametrically. In all of the analysis of the interactions we assumed a constant value of lambda which gave rise to the best qualitative approximation of the experimentally observed intertrial interactions. The choice of this parameter does not alter the conclusions. The models were fit using a pattern search algorithm and the parameters used in the pattern search are shown in Table 1. To investigate the robustness of our fitting procedure we fit synthetic data (please see response to reviewer 3 for additional details). These simulation results show that within the ranges of parameters that maximize the goodness of fit to the experimental data (measured using KL divergence), our fitting procedure produces good estimates of the parameters (Supplementary Figure 2). We also now include the investigation of the robustness of fit with respect to changes in the fit parameters (Supplementary Figure 6).*

3. The usefulness and meaning of the results reported in the last paragraph (lines 325-337) is unclear. It is confusing to report that discrete attractor models are better than the continuous one (but that the plasticity model is even better than the discrete model when delay time is considered) and then right below it report a seemingly contradicting result where the continuous model out performs the plasticity model. The authors are encouraged to motivate this analysis better in the context of the study aims and reconcile the interpretation of this result in respect to the other results reported in the study.

Our Response: *Thank you. We have rephrased the last section of the results to make the objective of the analysis and its conclusions clearer. The plasticity model has two components: the static landscape component and the plasticity component. Empirical fits to the data suggest that the static component does indeed have discrete energy wells. Also, consistent with previous results suggesting that the discrete energy wells stabilize the system against noise, a noisy discrete attractor system does produce more accurate energy*

performance than a line attractor system at longer delays (Figure 5E). This noise stabilizing property of the discrete attractor system has been used to explain why discrete, rather than continuous attractors, appear to be used by working memory systems. However, our results suggest that noise stabilization is accomplished, at least in part, through the plastic component of the model implemented as a local deformation in the energy landscape. Thus, it is unclear whether in a system with plasticity, the static discrete attractor component of the model offers any additional advantage over a continuous attractor system with plasticity. Results in Figure 5F show that this is not the case. Addition of a minimal amount of plasticity to a continuous attractor system is sufficient to stabilize the system. Because plasticity superimposed onto the continuous attractor system does not introduce bias, the overall memory storage in a continuous attractor system with plasticity is better than in a discrete attractor system with plasticity. Thus, noise-stabilizing properties of discrete attractor systems do not appear to explain why they appear to be used by working memory systems. An interesting alternative explanation for why discrete attractors are used by working memory is that they may allow for “chunking” of memories, as we suggest in the discussion. However, because our stimuli were deliberately made very simple, it is not possible to use our dataset to say anything specific about chunking. This suggestion could be addressed in the future work with more complex stimuli.

4. From the paper: “We find that regardless of the amount of plasticity added to the model the continuous attractor always outperforms the discrete attractor (p-value < 0.001).” This is a bit confusing, because in theory it would make sense that as the amount of plasticity approaches 0, the “plasticity” model should approximate the discrete attractor model. If it does not, does this suggest there were other changes added in the plasticity model (on top of just plasticity) that is not in discrete attractor model that may account for the results. Testing this (as plasticity goes to 0, the plasticity and discrete model have the same results) would be an important control case to make sure that all else is equal between the models.

Our Response: *We apologize for wording this awkwardly in the original submission. The results in Figure 5F show the behavior of continuous and the discrete attractor system with plasticity set to zero. In this case, at longer time delays, discrete attractor system yields better performance at longer time delays as would be expected given noise stabilization properties of discrete attractors.*

Minor Concerns

1. From the paper: “with the addition of noise the discrete attractor model beats out the continuous attractor model given sufficient time, p-value < 0.001(Figure 5E, blue and green lines)”. Is this p-value calculated using all time delays, or as the text suggests, only uses the longer time delay values (is the p-value calculated on the whole line or part of the green vs. blue line).

Our Response: *Sorry about this. As we now state clearly the benefit of the discrete attractor system in this case is only observed at long time scales, as would be expected.*

2. The authors report that 0.8 is the best intertrial values and since 0.8 is also the lowest value tested, there should be some values less than 0.8 that are tested. Some statistics should be performed on the curves to show which value is the best/why it was chosen.

Our Response: *We now clearly state in the text that our experimental paradigm does not directly allow us to estimate the time constant for decay of the plasticity. This is because the inter-trial delay was not systematically varied. We chose 0.8 (as is now clearly explained in the text) because it matched the bulk PI measurements observed experimentally. Note, however, that the choice of this value does not affect the shape of the PI but only the magnitude of interference. Thus PI will be more or less costly to the overall performance based on the frequency of trials.*

3. In Figure 5D, “Spatial scale of bias” is a confusing y-axis title. The figure legend is helpful in explaining that it was the difference between the max and min deviations (the red lines in A). Perhaps consider changing the axis title to “PI magnitude” (as referred to in the text) or to a name that more intuitively described the results reported there.

Our response: *Thank you. We corrected the y-axis label.*

4. There is no reference in the text for Figure 5F.

Our response: *Thank you. We have now added an explicit callout to this panel.*

Reviewer #2 (Remarks to the Author):

Brennan and Proekt used a computational modeling approach to test the attractor dynamics in working memory. They found that the discrete attractor model by itself could not fully explain human behavior in a spatial working memory task. When plasticity was introduced to the discrete attractor model, the model better accounted for working memory bias across different time points as well as inter-trial serial dependence. They conclude that maintenance of working memory content relies on a stable structure with a plasticity mechanism.

Overall, I thought that this was a well-motivated paper that addresses the temporal dynamics of working memory. I found the paper enjoyable to read, the methods appropriate, and the results carrying implications for the literature on the discrete attractor model. My comments are fairly minor in nature. For example, there is literature on serial dependence bias that could be related to the intertrial dynamics observed in the current study and speak to the nature of the bias. This and other points are detailed below.

Our Response: Thank you.

1. Mechanisms for working memory bias.

a. There may be multiple sources for working memory bias. Behavioral observations show that the direction of bias may be attractive or repulsive, which has been linked to different mechanisms (e.g., Chunharas et al., 2022; Czoschke et al., 2019). For example, Fritsche et al. (2017) have suggested that positive bias reflects attraction from the previous decision, whereas negative bias reflects neural adaption that repels current perception away from previous stimuli. I am wondering whether/how the model in the current paper may address these biases that happen at different levels (low-level perceptual vs. high-level decisional).

Our Response: Thank you for bringing this to our attention. We now discuss these works in the manuscript. We note that the study by Chundaras and Czoschke et al use rather different experimental paradigms from ours. In contrast, Experiment 1 in Fritsche et al is similar to ours in spirit. The data from this experiment reveal predominance of the attractive bias which was nicely approximated by a derivative of Gaussian (DoG) with some minor deviations at the edges. We note that our model yields precisely DoG-type interference. It is possible that holding multiple stimuli in memory such as in Chundaras and Cozschke may introduce more complex interference patterns between preceding and current trials. In principle a plasticity model with a different “plasticity kernel” ζ_p (Eq. 5) can explain both the attractive and the repulsive interactions. In our data the dominant effect was that of an attraction to previous stimuli. Furthermore, an assumption of a Gaussian type deformation of the energy landscape is a parsimonious one. This is why it was chosen in our model. Future work may want to investigate different shapes of the plasticity kernel in more sophisticated experimental paradigms.

b. The authors assumed that the attractive bias in human data is a result of plasticity. However, related to my previous point, this type of attractive bias may be more likely to reflect a decisional bias rather than a perceptual one. Is it possible that despite the apparent similarity between the human data and the model simulation, the two may be reflecting different mechanisms?

Our response: We do not make any distinction between perceptual and decisional influences as these are not directly discernable at the level of our experimental observations. We are simply concerned with determining whether a phenomenological drift-diffusion model is capable of reproducing experimental data in a visual working memory task. Our major finding in this regard is that it cannot and plasticity is required. Whether this “plasticity” reflects perceptual or more downstream decisional variables is not directly addressed in the manuscript. Nevertheless, guided by your suggestion and that made by Reviewer 3, we now included a discrete attractor system where the dynamics occur in two stages (“encoding” and “memory”). We allow the drift in these two stages to be different. This, however, does not rescue the performance of the discrete attractor model, because the ultimate problem for this model is that it cannot explain behavior at long delays.

c. Regarding the intertrial interactions, this type of proactive interference (also referred to as 'serial dependence' in other contexts) may last more than one trial and the bias has been reported to change its direction from attractive to repulsive on a longer time scale. For example, Fritsche et al. (2020) observed that the attractive bias has a long-lasting effect across a couple of trials, and interestingly the bias turned to be repulsive after 5 trials. Did the authors observe similar cross-trial dynamics in the current dataset? If so, could this pattern be accounted for by the plasticity model?

Our Response: *Thank you for pointing us to this interesting work. We did not observe any long lasting effects. It should be noted that because the locations of the target stimulus were chosen at random, in order to appropriately isolate the effect of a particular trial in a sequence of preceding trials, an exceptionally large dataset is required. Furthermore, our trials were broken down into sessions as the task can be rather tedious and we wanted to avoid the effects of fatigue. Thus, our results do not directly speak to the existence of long lasting interference effects.*

2. Although Figure 4 showed that the energy landscape model with added plasticity could nicely match the temporal dynamics of bias from human data, I am not certain whether the model also tracks the change in precision across time. Specifically, comparing 4D (1-second delay) with 4E (20-second delay), memory precision seems to be quite similar. In contrast, human data in Figure 2C and 2D showed a clearly higher precision at 1 second than at 20 seconds. It was not entirely clear to me how the precision was accounted for by the model. More details on this could be helpful.

Our Response: *We fit the model to best recapitulate the observed distribution of responses by minimizing the KL-divergence between model-based and experimentally observed distributions. Thus, the model is designed to recapitulate both the bias and spread of responses around this mean bias. It is true that the fit plastic model exhibits less precision at early time delays as compared to the experimental data. Future work could address how the choice of specific cost function (i.e. KL-divergence) effect the qualitative fits of the models. Regardless, the goodness of fit is reported in terms of R^2 (which is above 0.9 for the plastic model at short and long delays). Additionally we report goodness of fit in terms of Earth Movers distance. Finally, we compare the plastic and the static model using AIC and the newly added BIC. All these measures argue strongly for the fact that a plastic model is a good fit to experimental observations across different delays, and certainly outperforms the static model.*

3. Could the authors clarify why plasticity is accomplished through the local deformation landscape? How many deformation points could there be and is it always going to match the number of discrete attractors? Lastly, why would a depression predict an attractive inter-trial bias, rather than an adaptation-like effect?

Our Response: *The operations of the model are illustrated in figure 4B of the manuscript. As the memory trace traverses the energy landscape it depresses the energy landscape locally*

(the shape of the depression was assumed to be Gaussian as described above). We only studied a paradigm where a single stimulus had to be remembered. It is not a priori clear what should happen in tasks where multiple stimuli are simultaneously held in memory but it is possible to modify the model to have more than one deformation (one for each memory trace concurrently held in working memory). How well such model would approximate performance on multiple stimulus paradigm would have to be determined by future work. The plasticity has nothing to do with the number of discrete attractors. For instance, one can implement plasticity (e.g. Figure 5) in a continuous attractor model.

Minor:

1. It was mentioned a few times that "... human working memory at steady state reflects significantly higher information about the stimulus than would be expected from the static landscape model" [lines 238-240]. What exactly is the information that is not accounted for by the static landscape model?

Our Response: *The mutual information is computed according to equation 2. Specifically equation 2 compares the entropy of the joint distribution of target and response to the entropy of each of the marginals (target and response). In other words, mutual information computes the amount of knowledge about the target stimulus given the response. The fact that more mutual information is observed in the human data than that expected from an attractor model simply means that there is more information about the position of the stimulus than could be recovered from the performance of the human observers after a long delay than would be expected from an attractor model. As we show, the maximal number of bits of information recoverable from an attractor model at steady state is simply $\log(n)$ where n is the number of attractors.*

2. The simulation results in Figure 5B showed a symmetrical pattern where the mean bias was around 0 when the inter-trial distance was 0. However, the sinusoidal pattern from the human subjects showed a shift towards the left, such that the bias was positive at distance of 0. Why would one expect to see a pattern like this?

Our Response: *Thank you for pointing this out. It is true that the data is slightly offset from zero. This could be fit by allowing the plasticity kernel to have a non-zero offset relative to the present location of the stimulus at the expense of an additional parameter. We did not pursue this in the present manuscript. This offset is relatively minor in the data and was not well approximated by the model as we now state directly. Please note, however, that the model fitting procedure was entirely agnostic to inter-trial interactions. Given this, the model predictions of inter-trial interactions are quite good.*

3. In Figure 5E: the time scale ranges from 0 to 170 s for the No plasticity model, but 0-22 s for the Plasticity model. Is there any reason for this?

Our response: *The time scale is extended for the no plasticity case to demonstrate that eventually the discrete attractor system performs better in the presence of noise. As can be*

seen from the figure, it takes a long time for the benefit of the attractor model to manifest. This time scale can be altered by changing the amount of noise.

4. It may be easier for the readers to follow if the terminologies could be consistent throughout. For example, is the “energy landscape model” the same as the “energy well model”? and also the “line attractor” and “continuous attractor”?

Our response: Thank you. We standardized the terminology throughout the manuscript.

Reviewer #3

Biases in working memory tend to accumulate over time. This has led to proposals that working memory may be well-described as a dynamical system in which memories drift towards attractor states. Here, the authors propose that the energy landscape underlying this dynamical is itself dynamic, changing over the course of the trial due to short-term synaptic plasticity. They argue that this plastic model parsimoniously captures key features of both intra- and inter-trial phenomena.

Overall, the paper is conceptually interesting and the use of a single modeling scheme to address both single-trial dynamics and intertrial interference is innovative. However, I have some concerns with the interpretation of the results and I feel the manuscript could be improved with further analysis.

Our response: Thank you.

General Comments

1. It would be helpful to have a sense of how participants performed overall before diving into the details of the model and derived statistics. For example, what was the error distribution look like at each delay, and how variable was mean error across subjects? Is it possible to estimate the lapse rate (‘guessing’) in this linear space?

Our Response: Thank you for this very useful suggestion. We now include the distribution of errors at each delay time in the newly modified Figure 1. Detecting guessing on this task relies on choosing an admittedly arbitrary threshold. We chose to define “guessing” as a response that was at least 0.25 away from the target. Using this threshold, we rejected a total of 210 out of 17760 trials (approximately 1% of all the data).

2. The static model analyzed by the authors differs from previous efforts in two major ways. First, the model is fit to aggregate features of the data (error distributions or bias) using Monte Carlo methods, rather than fitting the data to single trials responses using a non-stochastic estimate of the response PDF. This has the potential to influence both the accuracy and precision of parameter estimation, and therefore the behavior of the best-fitting model. This concern could be addressed by demonstrating that the fitting

procedure can adequately recover a range of parameters used to generate simulated data.

Our Response: *Thank you. Indeed, our approach here was to use Monte Carlo methods to fit the distributions of responses as a function of target location and delay time using KL-divergence as a cost function. As you suggested, we now include a validation of this procedure (newly added Supplementary Figure 2). We simulated data using a drift diffusion model and recovered the parameters using our fitting approach. As can be seen in the figure, our estimates are reliable.*

3. Second, the authors account for baseline bias and variance in behavior (at a hypothetical zero-delay) by allowing the model to evolve for a fit number of baseline timesteps. However, previous work suggests that the values of drift and noise during such baseline periods differs substantially from that during memory, and accounting for this in the model results in superior performance. Therefore, this has the potential to substantially affect the model fits presented in Figure 2. This concern could be addressed by fitting separate values of drift and noise during baseline.

Our Response: *Thank you for this very insightful suggestion. To address your concern, we performed separate model fitting using a static landscape model where the drift strength and noise in the interval between 0 and t_0 were added as parameters to be fit independently by the model. The results of this fitting procedure are shown in the newly added Supplementary Figure 5. As can be seen in this figure, allowing for different dynamics during the baseline period does not dramatically alter the conclusions. The primary reason why the static landscape model fails is that it is unable to fit the data at long delays. This long term behavior is not strongly affected by the dynamics during the initial baseline period.*

4. Two different fitting procedures were applied to the static model, while one fit of the plastic model is shown in Figure 4. However, I couldn't find which method was applied here. It would be helpful to clarify this explicitly in Figure 4 or the related text.

Our Response: *We regret not spelling this out better in the original submission. All models were fit using the same Monte Carlo methods. The only difference is that the plastic model has two additional parameters that describe the growth rate and the spatial extent of the deformation. This is now more clearly stated in the methods. The static model was also fit to explicitly recapitulate the time constant for the development of biases. This was done because the original fits of the static model produced highly inaccurate estimates of the time constant for bias development (Figure 2). Because the plastic model fit both the distribution of the responses and the bias development kinetics rather well, we did not feel that fitting the plastic model just to bias development was indicated and did not pursue it.*

5. In the methods section it wasn't clear how exactly the AIC values for model comparison were computed – could the authors clarify? Additionally, since AIC is known to favor more complex models, it would be helpful to report the BIC values alongside

these for comparison. Or, perhaps preferably, reporting cross-validated fits instead of AIC/BIC given the impressive size of the data set.

Our Response: *We agree with your assessment and address it in the revised version of the manuscript by comparing the model fits using both AIC and BIC. Both measures converge on the fact that the plastic model is a better fit to the data. This conclusion is further strengthened by the fact that static model is unable to store enough information about the stimuli at long delays. This is a model-independent conclusion. In contrast, fits of the plastic model store approximately as much information as human subjects at long time delays after biases saturate.*

6. The authors make the strong claim in the abstract that ‘dynamics of human working memory violate predictions made by the discrete attractor theory’, but ultimately conclude that ‘in total, our results thus far agree with the finding that discrete, rather than continuous attractors likely contribute to the storage of working memory, but show that plasticity is required to fully capture the dynamics’. Clearly, these are very different statements, and the latter is much more consistent with the results as they currently stand.

Our response: *We agree that the claim was not well articulated in the abstract. We revised the abstract substantially to make the claims consistent with the data. Our results suggest that while discrete attractors (rather than continuous) attractors do indeed contribute to maintenance of working memory, they are insufficient. Plasticity is required in order to explain why, after biases saturate, the mutual information between the stimulus and the response is higher than what can be expected from a discrete attractor system.*

7. These results seems potentially complimentary to (though quite distinct from) those of Eissa and Kilpatrick (biorxiv, 2022). A brief consideration of this in the discussion might be interesting.

Our response: *Thank you for alerting us to this very interesting paper. We agree that Eissssa and Kilpatric are interested in very similar phenomena, but address them in a very different context. We now cite this manuscript in the discussion.*

Specific Comments

1. There appears to be a typo on line 266 – “[The] plastic landscape”?

Our response: *Thank you.*

7th Aug 23

Dear Dr Proekt,

Your manuscript titled "Plasticity and attractor dynamics in human visual working memory." has now been seen by our reviewers, whose comments appear below. In light of their advice I am delighted to say that we are happy, in principle, to publish a suitably revised version in Communications Psychology under the open access CC BY license (Creative Commons Attribution v4.0 International License).

We therefore invite you to revise your paper one last time to address a list of editorial requests. At the same time we ask that you edit your manuscript to comply with our format requirements and to maximise the accessibility and therefore the impact of your work.

EDITORIAL REQUESTS:

Related to the list of requests in the attached document, we ask to pay attention to the following key aspects: Once you have re-ordered the manuscript structure to: Introduction - Methods - Results - Discussion (rather than Introduction - Results - Discussion - Methods), you will need to revise your Methods and Results sections to ensure that key information about the paradigm and analysis is already conveyed in the Methods and then not unnecessarily duplicated.

As you revise the Results, please pay very close attention to our requirements for statistics reporting and interpretation (detailed in the checklist and on our webpage). We emphasize in particular the requirements for the reporting and interpretation of non-significant outcomes in NHST.

Your figures and text clearly support each other well in most instances, and some figures aptly serve display purposes, rather than insights into statistical comparisons. However, please ensure that the text is readable without constant reference to the Figures, especially where you describe the results of a statistical tests. The main text should mention the quantities/conditions/models that are entered into a respective analysis by name, and note the insights offered by the statistics, rather than referring the reader to their visual display or the visual display of this analysis.

SUBMISSION INFORMATION:

OPEN ACCESS:

Communications Psychology is a fully open access journal. Articles are made freely accessible on publication under a [CC BY license](http://creativecommons.org/licenses/by/4.0) (Creative Commons Attribution 4.0 International License). This license allows maximum dissemination and re-use of open access materials and is preferred by many research funding bodies.

For further information about article processing charges, open access funding, and advice and support from Nature Research, please visit <https://www.nature.com/commspsychol/article-processing-charges>

At acceptance, you will be provided with instructions for completing this CC BY license on behalf of all authors. This grants us the necessary permissions to publish your paper. Additionally, you will be asked to declare that all required third party permissions have been obtained, and to provide billing information in order to pay the article-processing charge (APC).

* DATA AVAILABILITY:

[link redacted]

Best regards,

Marike, on behalf of Eva Pool

Marike Schiffer, PhD
Chief Editor
Communications Psychology

Eva R. Pool, PhD
Editorial Board Member
Communications Psychology
orcid.org/0000-0001-5929-1007

REVIEWERS' COMMENTS:

Reviewer #1 (Remarks to the Author):

This is an interesting and well-motivated study and the authors addressed previous concerns in a satisfying manner. The ms was improved in clarity and motivation. The study contributes to the understanding of how information is stored in working memory and the important benefits of plasticity models vs attractor models in capturing that.

Rachel Rac-Lubashevsky

Reviewer #2 (Remarks to the Author):

The authors have done an excellent job addressing all of my concerns. I have no further comments. Congratulations.

Reviewer #3 (Remarks to the Author):

Thanks to the authors for their thoughtful and detailed reply. They have addressed my concerns and I recommend the manuscript without reservation. Congratulations on a great paper!